# Structure of the molecular bushing of the bacterial flagellar motor

Tomoko Yamaguchi [1,2,7], Fumiaki Makino [1,3,7], Tomoko Miyata [1], Tohru Minamino [1], Takayuki Kato [1,4✉] & Keiichi Namba [1,2,5,6✉]

The basal body of the bacterial flagellum is a rotary motor that consists of several rings (C, MS and LP) and a rod. The LP ring acts as a bushing supporting the distal rod for its rapid and stable rotation without much friction. Here, we use electron cryomicroscopy to describe the LP ring structure around the rod, at 3.5 Å resolution, from *Salmonella* Typhimurium. The structure shows 26-fold rotational symmetry and intricate intersubunit interactions of each subunit with up to six partners, which explains the structural stability. The inner surface is charged both positively and negatively. Positive charges on the P ring (the part of the LP ring that is embedded within the peptidoglycan layer) presumably play important roles in its initial assembly around the rod with a negatively charged surface.

[1] Graduate School of Frontier Biosciences, Osaka University, Suita, Osaka, Japan. [2] RIKEN Center for Biosystems Dynamics Research, Suita, Osaka, Japan. [3] JEOL Ltd., Akishima, Tokyo, Japan. [4] Institute for Protein Research, Osaka University, Suita, Osaka, Japan. [5] RIKEN SPring-8 Center, Suita, Osaka, Japan. [6] JEOL YOKOGUSHI Research Alliance Laboratories, Osaka University, Suita, Osaka, Japan. [7]These authors contributed equally: Tomoko Yamaguchi, Fumiaki Makino. ✉email: tkato@protein.osaka-u.ac.jp; keiichi@fbs.osaka-u.ac.jp

The bacterial flagellum is a motility organelle responsible for rapid movement of bacterial cells towards more desirable environments. The flagellum consists of three structural parts: the basal body working as a rotary motor, the filament as a screw propeller, and the hook as a universal joint connecting the filament to the motor[1,2]. The flagellar motor converts the electrochemical potential difference of cations across the cell membrane to mechanical work required for high-speed rotation with almost 100% efficiency[3], and the maximum rotation speed has been measured to be 1700 revolutions per second (rps)[4], which is much faster than that of the Formula One racing car engine.

The basal body consists of the C ring, MS ring, LP ring and rod (Fig. 1). The MS-C ring acts as a rotor of the flagellar motor. The MS ring composed of a transmembrane protein, FliF, is the core of the rotor and the initial template for flagellar assembly. The C ring consists of three cytoplasmic proteins, FliG, FliM, and FliN, is attached to the cytoplasmic face of the MS ring, and is essential for torque generation and switching regulation of the motor. The stator proteins, MotA and MotB, form a proton ($H^+$) channel in the cytoplasmic membrane to couple $H^+$ influx through the channel to torque generation. The cytoplasmic domain of MotA interacts with FliG located on top of the C ring to generate motor torque while the periplasmic domain of MotB binds to the peptidoglycan layer for the stator unit be anchored[5]. The rod is a rigid and straight cylindrical structure directly connected with the MS ring and acts as a drive shaft that transmits the motor torque to the filament through a flexible universal joint called the hook[6,7]. The rod is divided into two structural parts: the proximal rod composed of FliE, FlgB, FlgC and FlgF; and the distal rod formed by FlgG. The hook and filament are composed of FlgE and FliC, respectively. FlgG and FlgE are identical in the N- and C-terminal regions (39%) although the rod and hook have different

mechanical and physical properties. Such differences come from the tip of a long β-hairpin structure called "L-stretch", which makes the rod rigid and straight[8]. The LP ring is a bushing supporting the distal rod for its rapid and stable rotation without much friction and is composed of a lipoprotein, FlgH, and a periplasmic protein, FlgI. The L and P rings are embedded within the outer membrane and the peptidoglycan (PG) layer, respectively[1,2]. The LP ring is very stable against various chemical treatment[9,10] and remains in the cell envelope with its central pore sealed even after the ejection of the flagellar axial structure containing the rod, hook and filament from the cell body[11–15]. Its inner surface must be smooth to sustain high-speed rotation of the rod even up to around 1700 rps[4]. High-resolution structures of the distal rod by electron cryomicroscopy (cryoEM) and X-ray crystallography have revealed its outer surface being smooth and highly negatively charged[6,7], leading to a plausible hypothesis that the inner surface of the LP ring may also be negatively charged to generate electrostatic repulsive force to keep the rod rotating at the center of the LP ring to minimize friction between them. However, how the LP ring assembles around the rod against the repulsive force remains unclear.

The flagellar axial proteins for assembly of the rod, hook and filament are transported from the cytoplasm to the distal end of their growing structures by a specialized protein export apparatus within the MS-C ring. In contrast, FlgH and FlgI are synthesized as precursors with a cleavable N-terminal signal peptide and are translocated to the periplasm by the Sec translocon[16]. The N-terminal signal peptides are cleaved during the translocation across the cell membrane[17,18]. FlgI molecules assemble around the distal rod with the help of the FlgA chaperone to form the P ring[19,20]. The P ring tightly associates with the PG layer, allowing the LP ring to act as a molecular bushing. The N-terminal domain

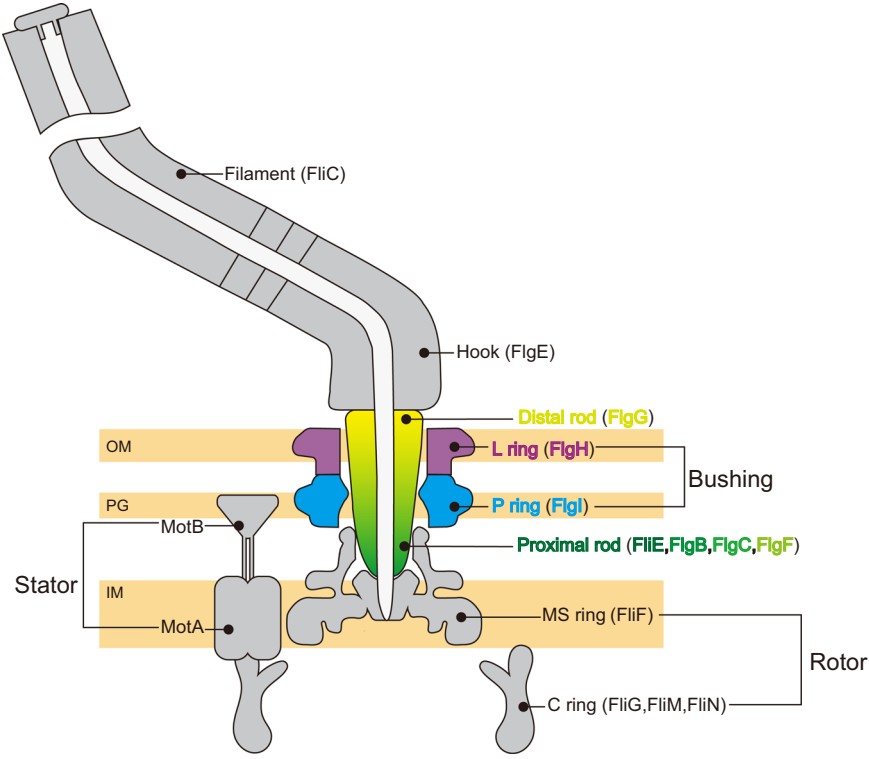

**Fig. 1 Schematic diagram of *Salmonella* flagellum.** The figure shows the central slice of the flagellum mostly in gray with only the rod and LP ring colored to highlight them. Component proteins forming each structural unit are written in brackets. The MS ring and C ring work as the rotor. The MotA/B complex acts as the stator, converting the energy of $H^+$ influx into motor rotation. The distal (yellow) and proximal (green) rods together work as a drive shaft transmitting motor torque to the hook and filament. The L ring (purple) and P ring (cyan) together act as a bushing and support the high-speed rotation of the flagellar motor. OM, PG and IM are the outer membrane, peptidoglycan layer and inner membrane, respectively.

of FlgI consisting of residues 1–120 is critical not only for the stabilization of FlgI but also for the formation of the LP ring[21]. Because the L ring is not formed in *flgI* null mutants[11,22], FlgH assembly around the rod presumably requires an interaction between FlgH and FlgI. Certain point mutations in the distal rod protein FlgG not only produce abnormally elongated rods called the polyrod but also allow many P rings to be formed around them, suggesting that the well-regulated P ring formation involves interactions of specific amino acid residues of FlgG and FlgI[23,24]. However, the detailed mechanism of LP ring formation remains elusive due to the lack of structural information.

To clarify the interactions between the rod and LP ring as well as the assembly mechanism of the LP ring, we carried out cryoEM structural analysis of the LP ring in the flagellar basal body from *Salmonella enterica* serovar Typhimurium (hereafter *Salmonella*). Here we report the LP ring structure at 3.5 Å resolution, showing 26-fold rotational symmetry and intricate intersubunit interactions of each FlgH subunit with six FlgH partners that explains the structural stability. The inner surface is charged both positively and negatively where positive charges on the P ring appear to play important roles in its initial assembly around the rod.

## Results

**Structure of the LP ring around the rod**. We purified the hook-basal body (HBB) complex from the *Salmonella* HK1002 cells (see Methods), collected cryoEM images, analyzed them by single particle image analysis using RELION[25], and analyzed the LP ring structure in the HBB. First, the three-dimensional (3D) image of the HBB was reconstructed at 6.9 Å resolution with C1 symmetry from 13,017 HBB images extracted from 12,759 cryoEM movie images (Supplementary Figs. 1 and 2). This density map contains the MS ring, LP ring, rod and proximal end of the hook. In the first 2D classification, only 1515 particles of 64,418 particles were classified as good images and others remained in the blurred classes. To collect more particles, another round of 2D classification was carried out with the blurred classes with the "ignore first peak" option in RELION[25], which enabled the collection of additional 15,902 particles. After 3D refinement from 13,017 particles, the C26 symmetry was visible in the P ring density and was confirmed by autocorrelation along the circumference (Supplementary Fig. 2b; see Methods for mor detail). In order to obtain a higher resolution structure of the LP ring, the LP ring part of the HBB image was re-extracted and analyzed with C26 symmetry, which produced a 3.5 Å resolution map (EMD-30398). To determine the relative positioning of the rod and LP ring precisely, a 3D image of the HBB was again reconstructed with C1 symmetry from 14,370 HBB images extracted with a larger box size from the same datasets, with the 3.5 Å resolution map of the LP ring used as a reference for the refinement, and this produced a 6.9 Å resolution density map (EMD-30409) with a better quality than the initial HBB map. Although the global resolution of this map was the same as that of the initial one, the local resolution was improved (Fig. 2a, Supplementary Fig. 2a; see Methods for more detail). The inner and outer diameters of the LP ring are 135 Å and 260 Å, respectively, and its height is 145 Å (Fig. 2a). The inner diameter of the LP ring is only slightly larger than the diameter of the rod (130 Å). There is an extra, rather blurred ring density beneath the P ring, which is likely part of FlgI (Fig. 2b). The position of the LP ring around the rod was determined by superimposing this high-resolution map on the HBB density map (Supplementary Fig. 3).

**Structures of FlgH and FlgI in the LP ring**. We built the atomic models of FlgH and FlgI based on the density map (PDB ID: 7CLR, Supplementary Movie 1). FlgH forms the L ring and FlgI

forms the P ring more or less exclusively (Fig. 2). The atomic model of FlgH forming the L ring contains the whole 211 residues of FlgH. The L ring adopts a three-layer structure composed of two β-barrel layers and the outermost layer with an extended chain (Fig. 2c, d and Supplementary Fig. 4). Two very long anti-parallel β strands, a short β strand and a short α helix make the inner layer domain, four anti-parallel β strands and two short α helices make the middle layer domain, and the N-terminal extended chain (outer layer chain) covers these two layers to form the outer layer. The anti-parallel β strands of the inner and middle layer domains are crossing nearly perpendicular to each other (Fig. 2d), forming a hydrophobic core with conserved residues (Supplementary Fig. 4a, c), and 26 copies of them form two layers of cylindrical β sheets in the inner and middle layers, respectively (Figs. 2c and 3a, b). The extended chain in the outer layer of the L ring is predicted to be disordered (Supplementary Fig. 4b), which is presumably disordered in the monomeric form of FlgH until it assembles into the L ring.

The atomic model of FlgI contains residues 1–125, 138–263 and 296–346, which cover 87% of the entire FlgI sequence (Fig. 2d, Supplementary Fig. 5). It is composed of three domains, which we named FlgI-IR$_U$ (upper inner ring), FlgI-IR$_L$ (lower inner ring) and FlgI-OR (outer ring). FlgI-OR (residues 1–11 and 160–230) is composed of three α helices and four β strands and forms the outer rim of the P ring. FlgI-IR$_U$ (residues 12–159) is composed of one α helix and 10 β strands, FlgI-IR$_L$ (residues 231–346) is composed of two α helices and five β strands, and these two domains form the inner part of the P ring. FlgI-IR$_U$ forms a β-barrel-like structure with highly conserved residues forming a hydrophobic core (Fig. 2d, Supplementary Fig. 5a, b) and also contains a loop of four polar residues from Thr-33 through Thr-36 (P ring loop in Fig. 2d). This loop and two highly conserved positively charged residues of FlgI-IR$_U$, Lys-63 and Lys-95, are in very close proximity to the rod surface (Fig. 2a), suggesting their involvement in P ring assembly around the rod as well as in the function as a bushing. FlgI-IR$_L$ is formed by the C-terminal chain of FlgI and contains a highly flexible loop (residues 264–295) that is likely to form the extra ring density beneath the P ring (Fig. 2b).

The flagellum and injectisome are both classified as the type III secretion system, but FlgH and FlgI are not homologous to the components of injectisomes[26], such as InvG of the type III secretion system[27], as well as those of other secretion systems: GspD of the type II secretion system[28], and PilQ of the type IVa pilus[29]. However, all of these proteins have similar cylindrical β barrel structures, except that the long β strands of FlgH shows different orientation (Supplementary Fig. 6). The structure of FlgI-OR looks similar to that of the N3 domain of GspD (residues 232–267, 288–323)[28], the N3 domain of InvG (residues 176–227, 252–302)[27], the N3 domain of PilQ (residues 251–321)[29], and the C-terminal domain of MotY (residues 155–202, 210–243, 262–271)[30], which shows an extensive similarity to the PG binding domain of OmpA family proteins interacting with the PG layer (Supplementary Figs. 6 and 7; analyzed by MATRAS[31]), suggesting the direct interaction of FlgI-OR with the PG layer. Because the LP ring is required for efficient transport of hook and filament proteins into the culture media, it also functions as a secretion-like pore-forming annulus in addition to that as a molecular bushing. So the structural and functional similarities of these proteins are probably the result of convergent evolution from different origins.

**Intersubunit interactions of FlgH and FlgI in the LP ring**. In the L ring structure, each FlgH subunit (H$_0$) interacts with six adjacent FlgH subunits (H$_{\pm1}$, H$_{\pm2}$ and H$_{\pm3}$) (Fig. 3a, b), and these

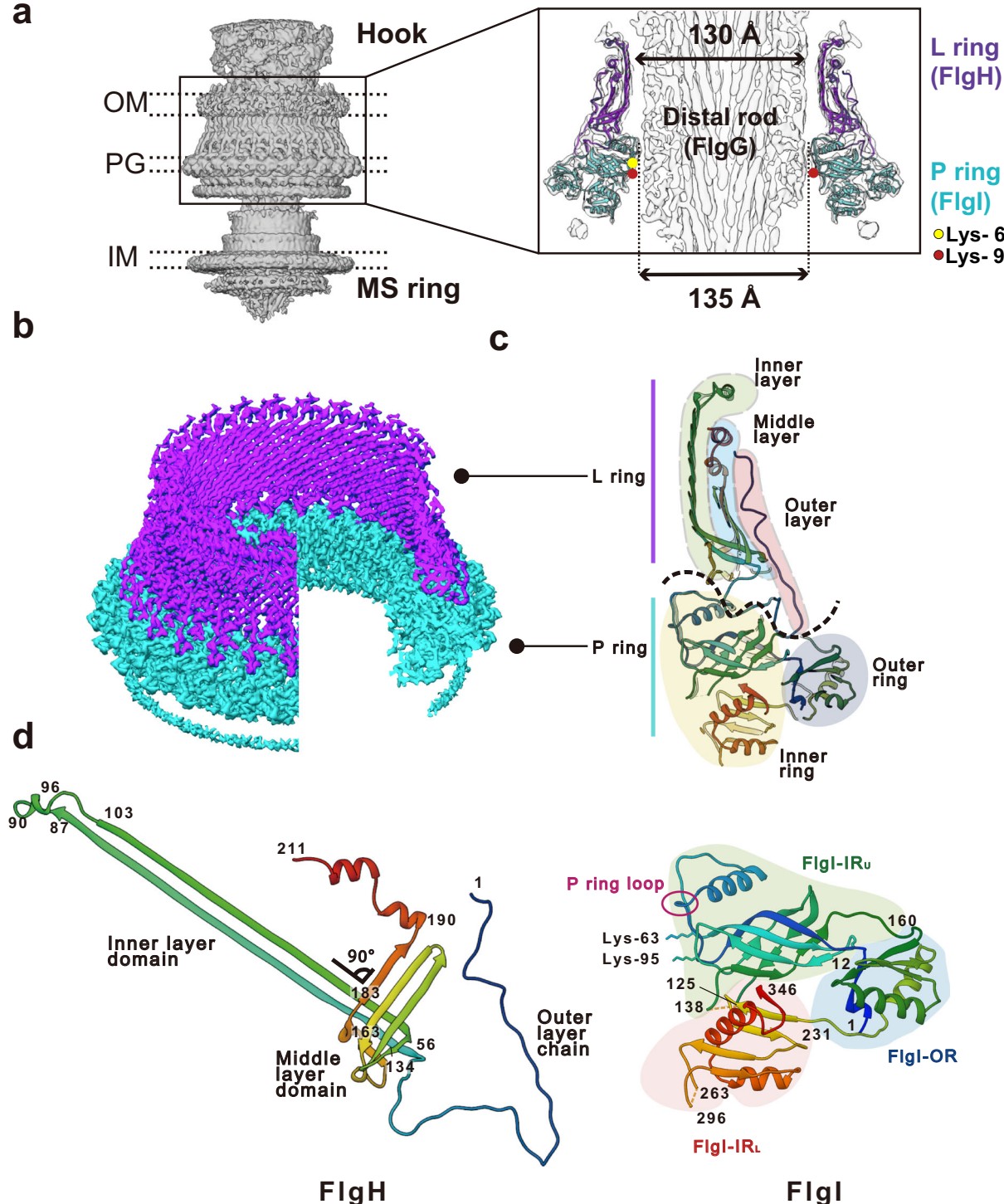

**Fig. 2 Density map and atomic model of the LP ring. a** Side view of the density map of the HBB complex (EMD-30409) without applying any symmetry (C1). OM, PG, IM refers to the outer membrane, peptidoglycan layer and inner membrane, respectively. A vertical slice of the LP ring and rod is shown enlarged on the right. The atomic models of the L ring (purple) and the P ring (cyan) are superimposed to the density map. The yellow and red circles indicate the positions of Lys-63 and Lys-95 of FlgI. **b** The density maps of the L ring (purple) and the P ring (cyan) after applying C26 symmetry (EMD-30398). **c** The atomic models of FlgH and FlgI in Cα ribbon representation in a vertical slice of the LP ring. The L ring has a three-layered structure (inner layer, light green; middle layer, cyan; outer layer, pink), and the P ring has a two-layered structure (inner ring, light yellow; outer ring, blue). **d** The atomic models of FlgH and FlgI (PDB ID: 7CLR) in Cα ribbon representation. FlgH is constituted from three parts: the inner layer domain; the middle layer domain; and the outer layer chain. FlgI is constituted from three domains: FlgI-OR (cyan); FlgI-IR$_U$ (light green); and FlgI-IR$_L$ (pink). Lys-63, Lys-95 and the P ring loop are located within the FlgI-IR$_U$ domain.

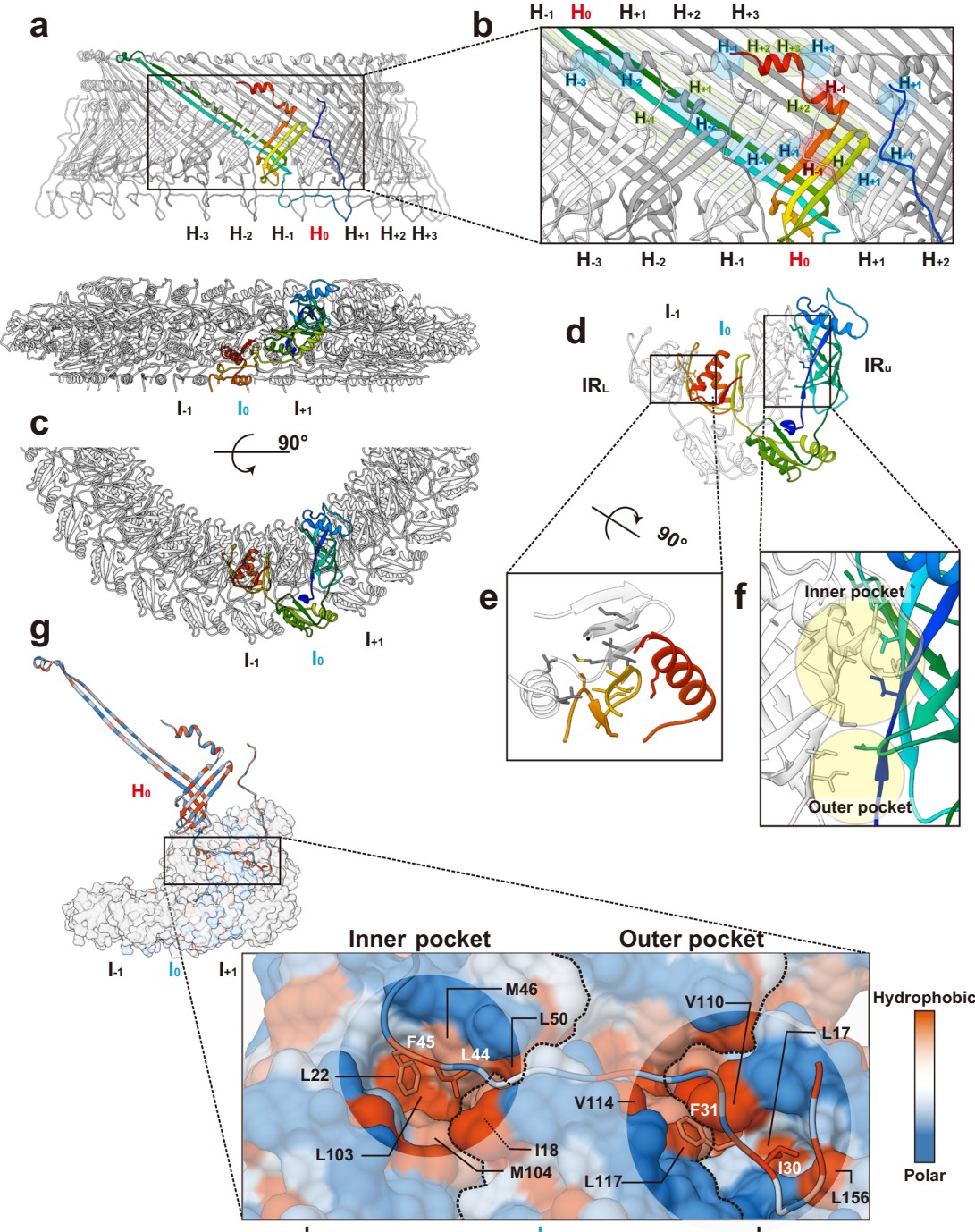

**Fig. 3 Intersubunit interactions of FlgH and FlgI in the LP ring. a** Side view of the L (upper panel) and P (lower panel) rings with one subunit colored rainbow. **b** Intersubunit interactions between a FlgH subunit ($H_0$) and its neighboring six FlgH subunits ($H_{\pm1}$, $H_{\pm2}$, $H_{\pm3}$). The segments of six neighboring FlgH subunits interacting with the subunit $H_0$ are labeled with their subunit ID and colored as: light green, interactions with the inner layer; light blue, interactions with the middle layer; and pink, interactions with the outer layer. **c**, **d** Top view of the P ring, showing the intersubunit interactions between FlgI ($I_0$) and its neighboring subunit ($I_{\pm1}$). **e** Hydrophobic interactions and a short anti-parallel β sheet formed by two neighboring FlgI-$IR_L$ domains. **f** Two hydrophobic pockets (inner and outer pocket) formed by two neighboring FlgI-$IR_U$ domains. Hydrophobic side chains are displayed in gray sticks in **e** and **f**. **g** Interactions between FlgH ($H_0$) and three FlgI ($I_0$, $I_{\pm1}$) subunits. The extended chain of FlgH in the outer layer form hydrophobic interactions with the inner and outer pockets formed by FlgI subunits. The position of I-18 of FlgI is indicated by a thin dashed line because it is behind Val-110. The boundary lines between FlgI subunits are indicated by thick dashed lines.

intricate intersubunit interactions make the LP ring mechanically stable. In the P ring, each FlgI subunit ($I_0$) interacts with two adjacent FlgI subunits ($I_{-1}$ and $I_{+1}$) (Fig. 3a, c, d). The FlgI-$IR_L$ domains of subunits $I_0$ and $I_{-1}$ interact with each other through hydrophobic interactions and also form a short anti-parallel β

sheet (Fig. 3e). FlgI-$IR_U$ forms two hydrophobic "pockets" namely inner and outer pockets (Fig. 3f). The C-terminal half of the outer layer chain of a FlgH subunit ($H_0$) lies at the bottom of the L ring and interacts with three FlgI subunits ($I_0$, $I_{\pm1}$) through its interactions with the hydrophobic inner and outer pockets of the P

ring (Fig. 3g). Leu-44 and Phe-45 of FlgH are in the inner pocket composed of Leu-22, Ile-18, Met-46, Leu-50, Leu-103 and Met-104 of FlgI (Fig. 2g left). In the outer pocket, Ile-30 of FlgH interacts with Leu-17 and Leu-156 of FlgI, and Phe-31 of FlgH interacts with Val-110, Val-114 and Leu-117 of FlgI (Fig. 2g right). The residues of these hydrophobic interactions, especially those in the inner pocket, are well conserved (Supplementary Figs. 5c and 8).

**Surface potential and conservation of the LP ring and rod**. In contrast to the prediction that electrostatic repulsive force by the negative charges on the surfaces of the rod and LP ring may play an important role in the bushing function, the inner surface of the LP ring is charged both negatively and positively. Asp-78, Asp-86 and Glu-104 of FlgH form a negative charge belt, and Lys-69 and Lys-114 of FlgH form a positive charge belt, suggesting that both repulsive and attractive force keep the negatively charged rod rotating at the center of the L ring (Fig. 4a). We also found that

Lys-63 and Lys-95, which are relatively well conserved among FlgI homologs, form a positive belt on the inner surface of the P ring (Fig. 4a, b) and these residues are in very close proximity to a negative charge cluster on the rod outer surface, raising a plausible hypothesis that these two lysine residues are critical for P ring assembly around the rod.

To examine whether these positive charges contribute to the P ring assembly, we replaced Lys-63 and Lys-95 with alanine or oppositely charged residue (Asp), constructed eight *flgI* mutants, *flgI*(K63A), *flgI* (K63D), *flgI*(K95A), *flgI*(K95D), *flgI*(K63A/K95A), *flgI*(K63A/K95D), *flgI*(K63D/K95A) and *flgI*(K63D/K95D), and analyzed their motility in 0.35% soft agar. We used a plasmid vector (pET22b, V) and a wild-type plasmid (pTY03, WT) (Supplementary Table 1) as negative and positive controls, respectively. The *flgI*(K63A), *flgI*(K95A) and *flgI*(K95D) mutants formed motility rings although not at the wild-type level (Fig. 5a, first row). The *flgI*(K63D) and *flgI*(K63A/K95A) mutants showed a very weak motility phenotype compared to the vector control (Fig. 5a, second row). The *flgI*(K63A/K95D), *flgI* (K63D/K95A) and *flgI*(K63D/K95D) mutants exhibited no motility

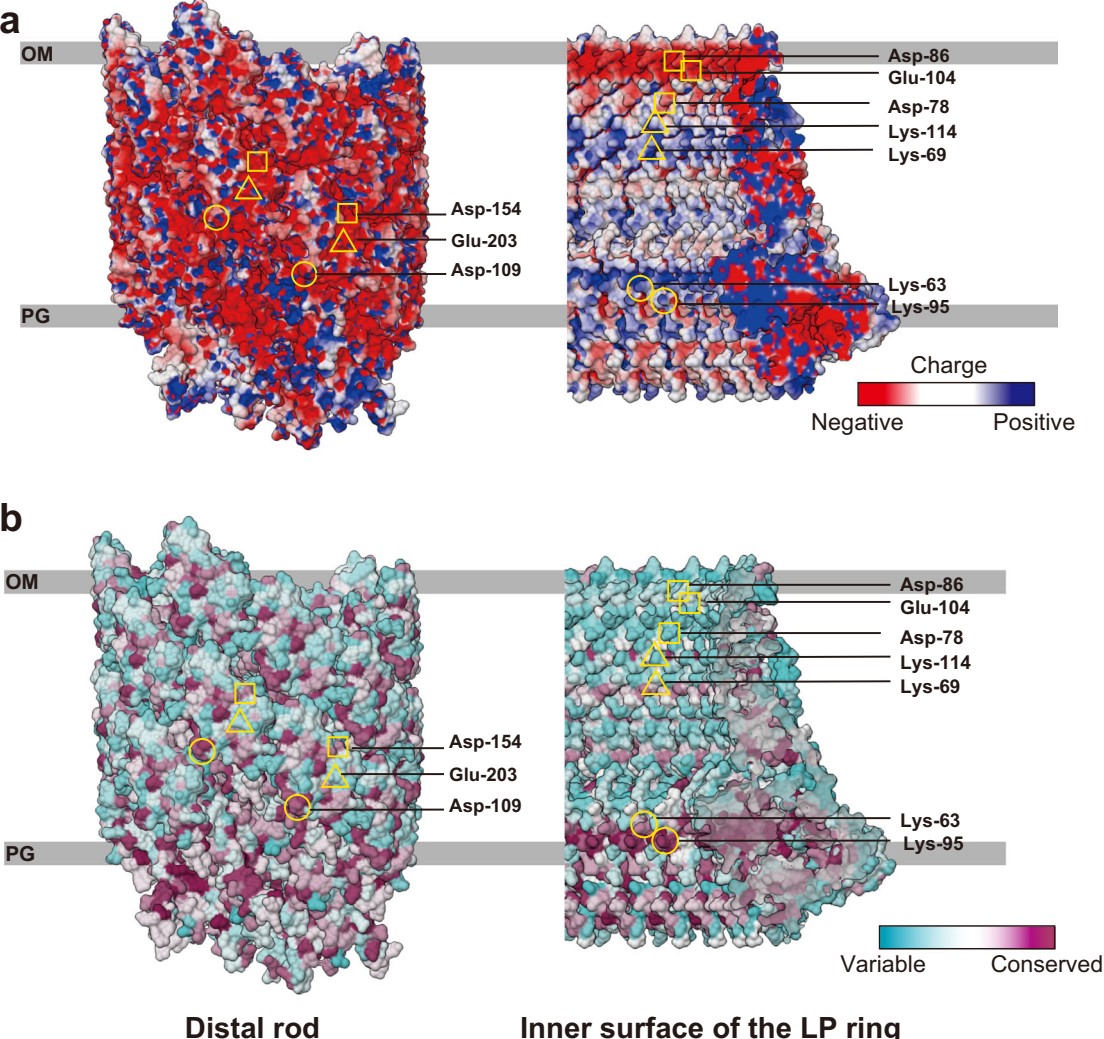

**Fig. 4 Electrostatic surface potential and sequence conservation of the rod and LP ring. a** Electrostatic potential of the outer surface of the distal rod and the inner surface of the LP ring. The helical arrangement of Asp-109 (circle), Asp-154 (square) and Glu-203 (triangle) of FlgG on the surface of the rod form negatively charged patches. Asp-86, Glu-104 and Asp-78 of FlgH (square) form a negatively charged belt on the inner surface of the L ring. Lys-69 and Lys-114 (triangle) of FlgH and Lys-63 and Lys-95 (circle) of FlgI form positively charged belts on the inner surface of the LP ring. **b** Conservation of amino acid residues on the outer surface of the distal rod and the inner surface the LP ring. Asp-109 (circle) and Aps-154 (square) of the rod and Lys-63 and Lys-95 (circle) of the P ring are highly conserved. The sequence conservation is colored from cyan to magenta (range of 0–100%), calculated and visualized by Chimera[50].

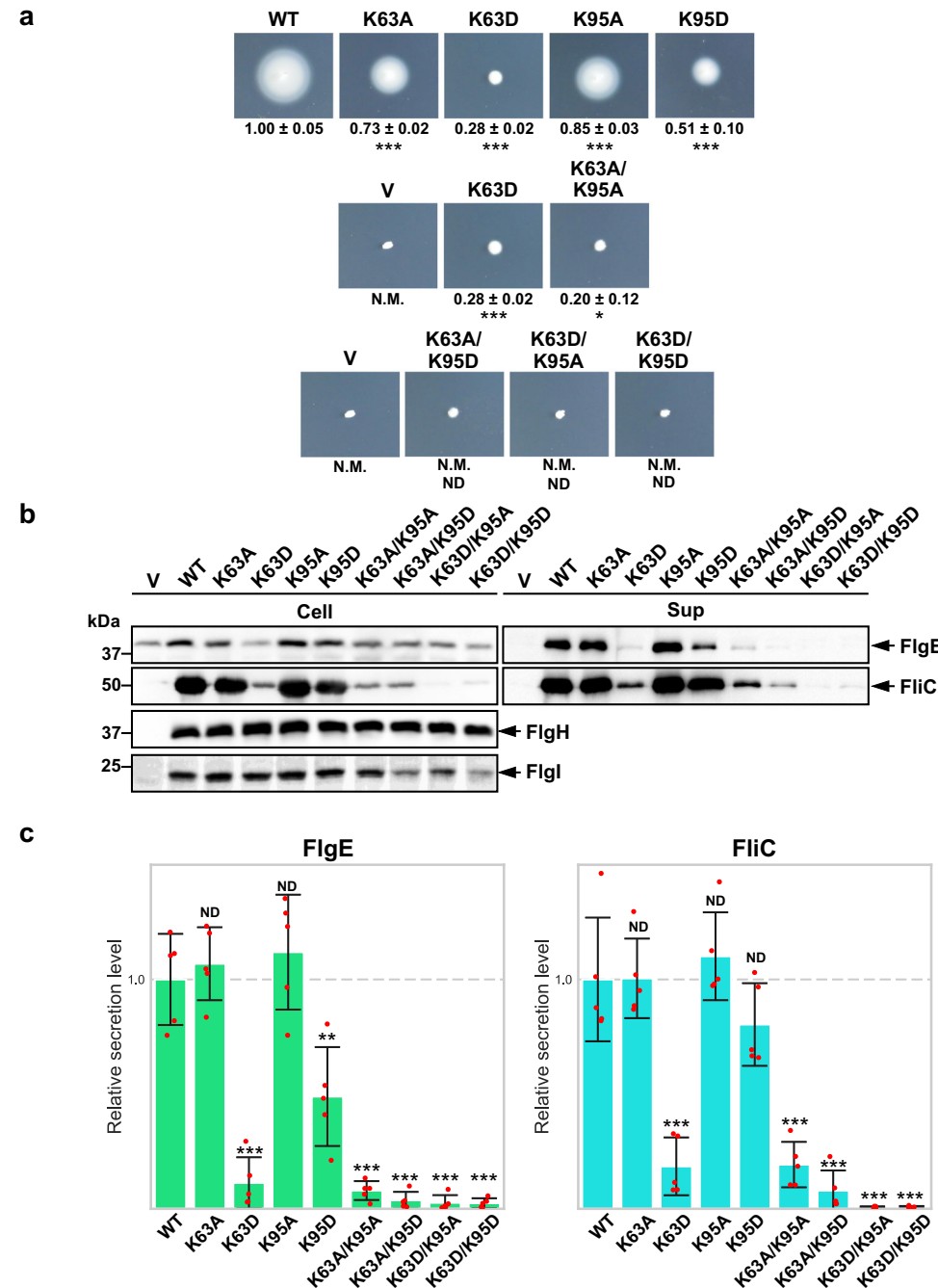

**Fig. 5 *flgI* mutation analysis by motility and secretion assay. a** Motility assay in 0.35% soft agar plate of the *flgI*(K63A), *flgI*(K63D), *flgI*(K95A), *flgI*(K95D), *flgI*(K63A/K95A), *flgI*(K63A/K95D), *flgI*(K63D/K95A) and *flgI*(K63D/K95D) mutants. N.M. indicates non-motile. The letter V and WT refers to the empty expression vector (pET22b) and wild type *flgI* expression vector (pTY03) (Supplementary Table 1) for the negative and positive controls, respectively. The diameter of the motility ring of five colonies of each strain was measured after incubating 8.5 h under 30 °C. The average diameter of the motility rings of the wild-type strain WT was set to 1.0, and then relative diameters of the motility rings of mutant cells were normalized to WT (mean ± SD, *n* = 5). The relative diameter of V was 0.16 ± 0.03. Comparisons between datasets were performed using a two-tailed Student's *t* test. A *P* value of <0.05 was considered to be statistically significant difference. *$P < 0.05$; ***$P < 0.001$; ND no statistical difference. **b** The cellular expression levels of FlgE, FliC, FlgH and FlgI (left) and the secretion levels of FlgE and FliC (right) by immunoblotting with polyclonal anti-FlgE, FliC, FlgH and FlgI antibodies. The positions of molecular mass markers (kDa) are shown on the left. The original immunoblots are shown in Supplementary Fig. 11 and then the contrast and brightness were adjusted using a software, Photoshop (Adobe). **c** Relative secretion levels of FlgE and FliC were quantified and presented in the bar diagrams (mean ± SD, *n* = 5). The average secretion level of the wild-type strain WT was set to 1.0. Comparisons between datasets were performed using a two-tailed Student's *t* test. A *P* value of <0.05 was considered to be statistically significant difference. **$P < 0.01$; ***$P < 0.001$; ND no statistical difference.

(Fig. 5a, third row) but showed some motility after prolonged incubation at 30 °C (Supplementary Fig. 9a). The flgI(K63A/K95D), flgI(K63D/K95A) and flgI(K63D/K95D) mutations affected the protein stability of FlgI, but other mutations did not at all (Fig. 5b). The flagellum is necessary for motility, and the disturbance of P ring assembly leads to the inhibition of flagellum formation and no motility. Since the swarm size and the number of the filaments were correlated (Supplementary Fig. 9b), these results indicate that the positive charges of Lys-63 and Lys-95 are both critical for FlgI to form the P ring.

The LP ring is required for the secretion of export substrates, such as the hook capping protein FlgD, the hook protein FlgE and the filament protein FliC, into the culture media[32]. In the absence of the LP ring, FlgD and FlgE can cross the cytoplasmic membrane but not the outer membrane, suggesting that the LP ring is required for forming the pore in the outer membrane to expose the distal end of the rod in the cell exterior to allow hook assembly outside the cell body. To test whether these FlgI mutations affect P ring assembly, we analyzed the secretion levels of FlgE and FliC by immunoblotting with polyclonal anti-FlgE and anti-FliC antibodies (Fig. 5b, c). The FlgE and FliC secretion by the flgI(K63A) and flgI(K95A) mutants were at the wild-type level. Because the swarm plate motility and the number of the flagellar filaments of these two mutants were significantly lower than that of the wild-type ($p < 0.001$ and $p < 0.01$, respectively), the K63A and K95A mutations probably inhibited P ring assembly by neutralizing their positive charges. The FlgE secretion by the flgI (K95D) mutant was significantly lower than the wild-type level ($p < 0.01$), suggesting that this mutation also affected P ring assembly by reversal of its positive charge. The K63D, K63A/K95A, K63A/K95D, K63D/K95A and K63D/K95D mutations considerably reduced the secretion levels, suggesting that these five mutations inhibited P ring formation. Although those five mutations showed no motility after incubation for 8.5 h (Fig. 5a), they showed some motility after incubation for 24 h (Supplementary Fig. 9a), which is consistent with the very small amounts of FliC and FlgE secretion (Fig. 5b, c). From these results, we propose that electrostatic interactions of Lys-63 and Lys-95 of FlgI with a negative charge cluster on the surface of the rod are essential for efficient P ring formation around the rod.

## Discussion

The atomic model of the LP ring around the rod provided many insights into the mechanical stability, bushing function, and assembly mechanism of the LP ring. The LP ring is stable even against a harsh treatment by acids and urea[9,10] and remains in the cell envelope even after the ejection of the flagellar axial structure from the bacterial cell body[11–15]. This extremely high stability can now be explained by the intricate intersubunit interactions between FlgH subunits, between FlgI subunits and between FlgH and FlgI. FlgH forms the three-layer cylinder structure of the L ring, consisting of two cylindrical β-barrels with their β-strands oriented nearly perpendicular to each other and the extended chain in the third layer lining them on the outer surface (Figs. 2c, d and 3a, b). Each FlgH subunit interacts with six adjacent FlgH subunits, making the L ring structure even more stable. Intersubunit interactions of FlgI in the P ring are not so extensive as those of FlgH in the L ring but each of FlgI three domains intimately interacts with domains of the adjacent subunits, even by forming intersubunit β-sheets (Fig. 3c–f). Then the extensive hydrophobic interactions between the extended chain of FlgH at the bottom surface of the L ring and the two pockets of FlgI on the upper surface of the P ring make the entire LP ring structure very stable (Fig. 3g).

The outer surface of the rod and inner surface of the LP ring are both very smooth and form a small gap between them that is large enough to accommodate one or two layers of water molecules in most part of their interfaces (Fig. 2a), indicating that these two structures are optimally designed for free rotation of the rod inside the LP ring without much friction. The outer surface of the rod is highly negatively charged[6,7] but the inner surface of the LP ring is charged both negatively and positively, probably to balance repulsive and attractive forces to keep the rod stably positioned and rotating at the center of the LP ring (Figs. 4a and 6a–e). Nanophotometric observations of individual motor rotation have shown 26 steps per revolution[33,34], and these data have been interpreted as the steps being generated by the repeated association and dissociation of the stator protein MotA and the rotor protein FliG for torque generation because the number of FliF-FliG subunits forming the MS ring and part of the C ring was thought to be 25 to 26 at the time[35–37]. However, recent structural analysis of the MS ring revealed its symmetry to be 33-fold with a variation from 32 to 35 (Ref. [38]), and the analysis of the flagellar basal body showed the MS ring to have 34-fold symmetry without variation and the C ring to have a small symmetry variation around 34-fold[39]. These results indicate the number of FliG on the C ring is mostly 34 and therefore invalidate the previous interpretation that the 26 steps per revolution reflect the elementary process of torque generation. Because the LP ring has 26-fold symmetry, the observed steps may be caused by potential minima formed by electrostatic force interactions between the rod and LP ring.

Although the LP ring is a stable component of the HBB even after its isolation from the cell, the LP ring easily slips off the rod when the basal body is isolated without the hook[9,22,39], indicating very weak interactions between the rod and LP ring to keep the LP ring in the proper position around the rod. In the absence of FlgH, however, the basal body formation stops just after completion of P ring assembly by FlgI around the rod to form a structure called "candlestick"[22,40], indicating that the P ring is tightly attached to the rod until the L ring is assembled above it. Among the flgG mutants that form the polyrod, many P rings are formed around the polyrod by flgG (G65V) and flgG (P52L) mutants whereas no P ring is observed by FlgG mutations of G53R, G183R or deletion of residues 54–57 (Ref. [23,24]). All of these FlgG mutations are located in the long β-hairpin structure called "L-stretch" formed by Salmonella FlgG specific sequence not present in FlgE (Supplementary Fig. 10a, b), and its tip is presumably exposed on the surface of the rod[7,41,42] (Supplementary Fig. 10c–e). Our model of the LP ring shows that Lys-63 and Lys-95 of FlgI are located on the inner surface of the P ring and directly face the negatively charged surface of the rod. Their mutations to either alanine to diminish the positive charge or aspartate to have the negative charge impair cell motility possibly by inhibiting P ring formation (Fig. 5). Hence, these two lysine residues are likely to play important roles in P ring assembly on the negatively charged surface of the rod. This is consistent with no P ring assembly by FlgG polyrod mutations G53R and G183R, which would disturb P ring assembly by repulsive force (Supplementary Fig. 10c), as well as deletion of restudies 54–57, which would put Glu-58 of FlgG at the tip of the L-stretch too far from Lys-63 and Lys-95 of FlgI (Supplementary Fig. 10d). We therefore suggest that Lys-63 and Lys-95 of FlgI are essential for the initial binding of FlgI to the rod surface to form the P ring tightly attached to the rod and that some conformational changes occur in the P ring upon binding of FlgH for L ring assembly to make the completed LP ring free from the rod for its free, high-speed rotation within the LP ring as a bushing (Fig. 6f). Structural analysis of the P rings stably attached to the polyrod is underway to clarify the mechanism of LP ring assembly. Understanding the mechanical and dynamic properties of the LP ring as a nanoscale bushing would also be quite interesting in physics and

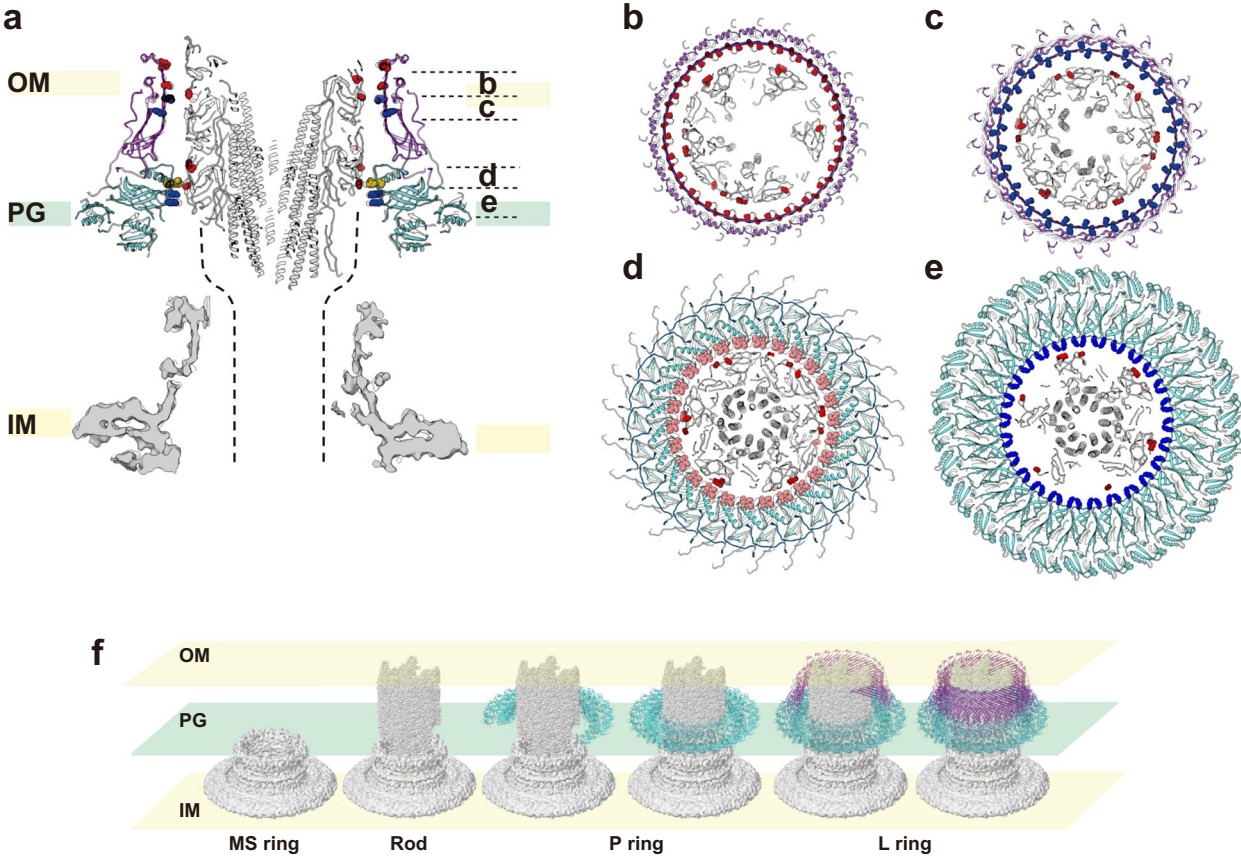

**Fig. 6 Interactions between the rod and LP ring and their assembly process. a** Vertical section of the atomic model of the rod (gray), L ring (purple), P ring (cyan) with the map of the MS ring at the bottom (gray). Charged amino acid residues are displayed with negative in red (Asp-109, Asp-154, Glu-203 of FlgG and Asp-78, Asp-86, Glu-104 of FlgH) and positive in blue (Lys-69, Lys-114 of FlgH and Lys-63, Lys-95 of FlgI). Four polar residues from Thr-33 to Thr-36 of FlgI in the P ring loop, which is located most closely to the rod surface, are shown in pink. The dashed lines are where the proximal rod is expected to exist. Axial view of four horizontal slices of the rod and LP ring as indicated in **a**, showing repulsive and attractive interactions between the rod and LP ring: **b** repulsive interactions; **c** attractive interactions; **d** the smallest gap between the rod and LP ring with non-charged residues of FlgI in the P ring loop; **e** attractive interactions. **f** The assembly process of the rod (gray) and LP ring (purple and cyan).

useful in nanotechnology applications, but it requires fully atomic molecular dynamics simulations of the rod rotation within the LP ring. Although it would need to deal with tens of millions of atoms and therefore is not a simple task, such study is also underway because it is now feasible by the development of high-speed computers, such as Fugaku.

## Methods

**Bacterial strains, media, and bacterial growth**. Bacterial strains and plasmids used in this study are listed in Supplementary Table 1. For cryoEM structural analyses, a strain expressing only the hook-basal body and not the filament (HK1002) was used. The L-broth [LB, 1% (w/v) tryptone, 0.5% (w/v) yeast extract, 0.5% (w/v) NaCl] was used for culturing bacteria. Soft agar plate [1% (w/v) Bacto tryptone, 0.5% (w/v) NaCl and 0.35 % (w/v) Bacto agar] was used for swarming assay[43]. Ampicillin was added to LB at a final concentration of 100 μg/mL.

**DNA manipulation**. All primers used in this study were listed in Supplementary Table 2. The *flgHI* genes were amplified by TaKaRa Ex Taq® DNA polymerase (TaKaRa Bio) using the chromosomal DNA of the *Salmonella* wild-type strain SJW1103 as a template and primers, FlgHI_NdeI_Fw_2 and FlgHI_BamHI_Rv_3. The PCR products were digested with NdeI and BamHI and cloned into the NdeI and BamHI sites of the pET22b vector (Novagen) to generate a plasmid, pTY03. In order to clarify whether the positively charged Lys-63 and Lys-95 of FlgI is important for forming the P ring upon interacting with negatively charged surface of FlgG, each or both Lys-63 and Lys-95 were replaced by Ala or Asp residue. Site-directed mutagenesis was carried out using primer STAR® MAX DNA polymerase (TaKaRa Bio). The mutated genes were amplified by PCR using pTY03 as a template and pairs of complementary primers containing a mutagenized codon listed in Supplementary Table 2, and then the plasmids were introduced to *E. coli* NovaBlue cells (Novagen) for transformation.

**Hook-basal body (HBB) purification**. The *Salmonella* HK1002 cells were pre-cultured in 30 mL LB with overnight shaking at 37 °C and inoculated into a 2.6 L of fresh LB. The cells were grown until the optical density had reached an OD₆₀₀ of about 1.0. The cells were collected by centrifugation and resuspend in 10% sucrose buffer [10% (w/v) sucrose, 0.1 M Tris-HCl, pH 8.0]. EDTA (pH 8.0) and lysozyme were added to final concentrations of 10 mM and 1.0 mg/mL, respectively. The cell lysates were stirred on ice for 1 h at 4 °C, and 0.1 M MgSO₄ and 10% (w/v) Triton X-100 were then added to final concentrations of 10 mM and 1% (w/v), respectively. After stirring for 1 h at 4 °C, 0.1 M EDTA (pH 11.0) was added to a final concentration of 10 mM. The solution was centrifuged at 15,000 × *g*, and the supernatant was collected. The pH was adjusted to 10.5 with 5 N NaOH and recentrifuged at 15,000 × *g* to remove undissolved membrane fractions. The supernatant was centrifuged at 67,000 × *g* to collect HBBs as a pellet. This pellet was resuspended in 1 mL of Buffer C [10 mM Tris-HCl, 5 mM EDTA, 1% (w/v) Triton X-100] and was centrifuged at 9700 × *g*, and the supernatant was collected. HBBs were purified by a sucrose density-gradient centrifugation method with a gradient of sucrose from 20 to 50%. Fractions containing HBBs were collected and checked by SDS-PAGE with Coomassie Brilliant Blue staining. The fractions containing HBBs were two-fold diluted by a final buffer [20 mM Tris-HCl, pH 8.0, 150 mM NaCl, 0.05% (w/v) Triton X-100], and the sucrose was removed by centrifugation at 120,000 × *g*. Finally, HBBs were resuspended with 10–50 μL of the final buffer, and the sample solution was stored at 4 °C.

**Electron cryomicroscopy and image processing**. A 3 μL sample solution was applied to a Quantifoil holey carbon grid R1.2/1.3 Mo 200 mesh (Quantifoil Micro Tools GmbH, Großlöbichau, Germany) with pretreatment of a side of the grid by 10 s glow discharge. The grids were plunged into liquid ethane at a temperature of liquid nitrogen for rapid freezing[44] with Vitrobot Mark IV (Thermo Fisher Scientific) with a blotting time of 5 s at 4 °C and 100% humidity. All the data collection was performed on a prototype of CRYO ARM 200 (JEOL, Japan) equipped with a thermal field-emission electron gun operated at 200 kV, an Ω-type energy

filter with a 20 eV slit width and a K2 Summit direct electron detector camera (Gatan, USA). The 12,759 dose-fractionated movies were automatically collected using the JADAS software (JEOL). Using a minimum dose system, all movies were taken by a total exposure of 10 s, an electron dose of 0.9 electrons/Å$^2$ per frame; a defocus range of 0.2–2.0 μm, and a nominal magnification of 40,000×, corresponding to an image pixel size of 1.45 Å. All the 50 frames of the movie were recorded at a frame rate of 0.2 s/frame. The defocus of each image was estimated by Gctf-v1.06 (Ref.[45]) after motion correction by RELION-3.0-β2 (Ref.[25]). The HBB particle images were automatically picked by an in-house python program (YOLOPick) that utilizes a convolutional neural network program, YOLOv2 (Ref.[46]). In total, 64,418 particles were extracted in a box of 340 × 340 pixels and used for 2D classification by RELION 3.0-β2. First, clear 2D classes were selected (1515 particles), and the "ignoring CTF first-peak" option was applied for the remaining blur 2D classes (62,841 particles) to select additional 15,902 particles, and in total, 17,363 particles were selected. The 15,242 particles were selected from those 17,363 particles after re-centering their positions by in-house program named TK-center.py, then the 14,975 particles were selected after another 2D classification, and those particles were used for first 3D classification into two classes with C1 symmetry. A density map of better quality, reconstructed from 13,017 particles (87%), was then used as a reference for density subtraction to leave only the LP ring. The C26 symmetry was visible in the LP ring density map, but its cross-section image was transformed from the Cartesian to the polar coordinates, and its autocorrelation function was Fourier transformed to confirm the rotational symmetry to be 26-fold (Supplementary Fig. 2b). This estimation was carried out by in-house program named Tksymassign.py. Then, the C26 symmetry was applied for further image analysis to obtain a higher resolution 3D map, which resulted in 3.5 Å resolution at a Fourier shell correlation (FSC) of 0.143 after 3D refinement and CTF refinement procedure from 10,802 particles (EMD-30398) (Supplementary Figs. 1 and 2). In order to obtain detailed information on the relative positioning of the rod and LP ring without any symmetry (C1), 14,975 HBB particle images were extracted with a lager box size (400 × 400 pixels) from the same movie datasets described above and by using the 3.5 Å resolution LP ring map as a reference for refinement, 14,370 particles were selected after 3D classification and 3D refinement. Because the resolution was not improved by this 3D classification and 3D refinement, 2D classification and centering were also performed on this dataset. In order not to be affected by the hook density, the reference and mask without the hook density was used for next 3D classification and 3D refinement. A density map of 6.9 Å resolution (at the FSC of 0.143) was obtained with C1 symmetry after 3D refinement and CTF refinement procedure from 14,370 particles (EMD-30409). Although the global resolution of this map was same as the initial HBB map, the local resolution was improved. Summary of data collection and image analysis is described in Supplementary Table 3.

**Model building**. The atomic model of FlgH and FlgI was constructed by COOT (Crystallographic Object-Oriented Toolkit)[47], and PHENIX-1.13-2998 was used for auto sharpening of the cryoEM LP ring map and real-space refinement[48]. The secondary structures were predicted by PSIPRED[49]. In order to identify the minimum units composing L ring and P ring, only the Cα atoms were placed on the cryoEM map by COOT to make the main chain model of FlgH and FlgI. For further refinement, the densities around 4 Å from the Cα atoms were extracted from the LP ring density map, and the model was automatically refined by PHENIX real-space refinement and manually fixed by COOT. The side chains were added manually by COOT based on the cryoEM map and PSIPRED prediction. In order to consider the interactions between adjacent molecules, seven FlgH and three FlgI molecules were used for further refinement, and the central models of FlgH and FlgI were extracted to be the final model. The entire LP ring model was made by UCSF Chimera[50] using the C26 symmetry. The surface potentials were calculated by APBS[51] and PDB2PQR[52] and multi sequence alignment (conservation) was done by Clustal Omega[53]. All the figures of 3D maps and atomic models used in this paper were prepared by UCSF Chimera[50]. Summary of model refinement and statistics is described in Supplementary Table 3.

**Motility assays and counting the flagella number**. We transformed *Salmonella* SJW203 cells (Δ*flgH-flgI*) with a pET22b-based plasmid encoding FlgH and FlgI (pTY03) or its mutant variants. Fresh transformants were inoculated onto soft agar plates [1% (w/v) tryptone, 0.5% (w/v) NaCl, 0.35% Bacto agar] with 100 μg/mL ampicillin and incubated at 30 °C for 8.5 h (Fig. 5) and 24 h (Supplementary Fig. 9). Five to eight independent measurements were performed. Statistical analyses were done using KaleidaGraph (Hulinks) and Excel (Microsoft). Comparisons between datasets were performed using a two-tailed Student's *t* test. A *P* value of <0.05 was considered to be statistically significant difference (Supplementary Table 4). For counting the number of flagellar filaments per cell, the mutants described above were cultured in 5 mL LB with shaking overnight, 1 mL of the culture media was centrifuged, and cells were collected as pellet. The cells were suspended in water, stained by 0.2% phosphotungstic acid on a thin carbon-coated, glow-discharged copper grid and observed by a JEM-1011 transmission electron microscope (JEOL, Akishima, Japan) operated at an accelerating voltage of 100 kV. For each of the mutant strains, the number of filaments per cell was counted for 100 cells manually. The statistical analyses were done using Excel (Microsoft). Comparisons between

datasets were performed using a two-tailed Student's *t* test. A *P* value of <0.05 was considered to be statistically significant difference (Supplementary Table 4).

**Secretion assays**. *Salmonella* SJW203 cells (Δ*flgH-flgI*) harboring a pET22b-based plasmid encoding FlgH and FlgI (pTY03) or its mutant variants were grown overnight in 5 mL LB with gentle shaking at 30 °C. A 50 μL aliquot of each overnight culture was inoculated into 5 mL LB and incubated at 30 °C with shaking until the cell density had reached an OD$_{600}$ of 1.4–1.6. Cultures were centrifuged to obtain cell pellets and culture supernatants. The cell pellets were resuspended in a sample buffer solution [62.5 mM Tris-HCl, pH 6.8, 2% sodium dodecyl sulfate (SDS), 10% glycerol, 0.001% bromophenol blue] containing 1 μL of 2-mercaptoethanol. Proteins in the culture supernatants were precipitated by 10% trichloroacetic acid and suspended in a Tris/SDS loading buffer (one volume of 1 M Tris, nine volumes of sample buffer solution)[54] containing 1 μL of 2-mercaptoethanol. Both whole cellular proteins and culture supernatants were normalized to a cell density of each culture to give a constant cell number. After boiling proteins in both whole cellular and culture supernatant fractions at 95 °C for 3 min, these protein samples were separated by SDS–polyacrylamide gel (normally 12.5% acrylamide) electrophoresis (SDS–PAGE) and transferred to nitrocellulose membranes (Bio-Rad) using a transblotting apparatus (Hoefer). Then, immunoblotting with polyclonal anti-FlgE, anti-FliC, anti-FlgH or anti-FlgI antibody (Medical & Biological Laboratories Co., LTD.) with 400-fold dilution was carried out using iBand Flex Western Device (Thermo Fisher Scientific) as described in the manufacturer's instructions. Anti-Rabbit IgG, HRP-Linked Whole Ab Donkey (GE Healthcare, Cat#SE250-10A-.75) with 1,000-fold dilution was used as the secondary antibody. Detection was performed with Amersham ECL Prime western blotting detection reagent (Cytiva). Chemiluminescence signals were captured by a Luminoimage analyzer LAS-3000 (GE Healthcare). The band intensity of each blot was analyzed using an image analysis software, CS Analyzer 4 (ATTO, Tokyo, Japan). All image data were processed with Photoshop software (Adobe). Five independent measurements were performed. Comparisons between wild-type and *flgI* mutant strains were performed using a two-tailed Student's *t* test. A *P* value of <0.05 was considered to be statistically significant difference (Supplementary Table 4).

**Reporting summary**. Further information on research design is available in the Nature Research Reporting Summary linked to this article.

## Data availability

The atomic model and density map of the LP ring have been deposited in the PDB database under accession code 7CLR and EMDB data base under accession code EMD-30398. The HBB density map has been deposited in the EMDB database under accession code EMD-30409. All the other data supporting the findings of this study are available within the article and its Supplementary Information and Source Data file. A reporting summary for this article is available as a Supplementary Information file. Source data are provided with this paper.

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

## Acknowledgements

We thank Kelly Hughes for providing strain SJW203, Hideyuki Matsunami for the HK1002 strain, Miki Kinoshita for help in genetic engineering and biochemical assay and for providing plasmids and Katsumi Imada for help and advice in constructing the atomic model. This work has been supported by JSPS KAKENHI Grant Number JP25000013 (to K.N.), JP18K06155 (to T.Miy.), JP26293097 and JP19H03182 (to T. Min.), and MEXT KAKENHI Grant Number JP15H01640 and JP20H05532 (to T.Min.). This work has also been supported by Platform Project for Supporting Drug Discovery and Life Science Research (BINDS) from AMED under Grant Number JP19am0101117 to N.K., by the Cyclic Innovation for Clinical Empowerment (CiCLE) from AMED under Grant Number JP17pc0101020 to K.N. and by JEOL YOKOGUSHI Research Alliance Laboratories of Osaka University to K.N. T.Y. was supported by RIKEN as a Junior Research Associate.

## Author contributions

T.K. and K.N. designed the project; F.M, T.K. T.Min., and K.N. designed the experiments; T.Y., F.M., T.Miy., and T.Min. prepared the samples; T.K. set up the cryoEM imaging and analysis system; T.Y., F.M., T.Miy., T.K. collected and analyzed cryoEM image data and built the atomic model; T.Y. and T.Min. performed genetic, biochemical and physiological experiments; all authors studied the atomic model; T.Y. and K.N. wrote a draft of the paper, and all the other authors discussed and contributed to writing up the paper.

## Competing interests

The authors declare no competing interests.
