## [Peer Review File · Nature Communications]

REVIEWER COMMENTS

Reviewer #1 (Remarks to the Author):

This is an elegant structural analysis of the LP ring of the flagellar motor. This cryo-EM revolution is great, it might be time at last to really understand the flagellar motor. Within this structure, the flagellar driveshaft rotates at speeds of hundreds to a couple of thousand cycles per second. It's truly mind-boggling. The detailed analysis of inter-subunit interactions helps to explain the incredible stability of this bushing and resistance to harsh chemical treatments. The structure also solves a prior mystery regarding differences in polyrod alleles of flgG that either do or do not (G53R & G183R) allow P-ring formation. I wonder if K63G and K95G substitutions in FlgI could suppress this (I am not requesting this just thinking about the possibility). One important dilemma is the fact that the P-ring must initially form around the FlgG distal rod structure, but the separate from the rod upon completion of the PL-ring structure. It's a beautiful structure

Reviewer #2 (Remarks to the Author):

Report for "Structure of the molecular bushing of the bacterial flagellar motor"

Reviewer: Mohammed Kaplan

This nice manuscript by (Tomoko Yamaguchi, Fumiaki Makino et al.) presents the first structure of the P- and L-rings of the flagellar motor at a resolution of 3.5 Å using single particle reconstruction cryo-EM. This structure allowed the authors to present a model of how these two rings might assemble and function as a bushing during the motor rotation. In recent months, high-resolution structures of various parts of the motor have been published, so this work is timely and helpful for the field. The manuscript (as is usually the case with the work done by Prof. Namba and colleagues) is a solid piece of work that deserves to be published in a prestigious journal like Nature Communications.

I have the following suggestions to the authors:

1- Recently, a manuscript from the lab of Prof. Susan Lea at Oxford (Johnson et al., see ref. (1)) was posted on the bioRxiv reporting the structure of the flagellar PL-rings (also from Salmonella) with other parts of the basal body like the rod and MS-ring. However, in their structure they saw that there is an extra density surrounding the FlgH density and after doing proteomics they identified it as (YecR). It is interesting that Yamaguchi and colleagues do not see an extra density around FlgH in their structure. Do they have an explanation for this? Johnson et al. hypothesize that YecR might remodel the lipids to help the assembly of FlgH (L-ring) because the lipid bilayer surrounding FlgH is shorter than usual in their structure. Can the authors here check the lipid bilayer thickness around FlgH in their structure? Is it also thinner? Probably the YecR ring disassembles at a certain stage after FlgH assembly? or do the authors have another explanation (hypothesis) for its absence from their structure? I think adding something (could be few sentences) regarding this point would be helpful for the reader.

2- The authors indicate that the P-ring (which assembles before the L-ring) undergoes a conformational change after the assembly of the L-ring around the rod (lines 268-272). Do they also think, based on their structure, that the P-ring is located closer to the MS-ring before the L-ring assembles as suggested by Johnson et al.? probably the authors can check for that if they already have a subset of particles that only have the P-ring? If not, the authors can probably indicate in Fig. 4 f that somehow the P-ring is different (in whatever way the authors find suitable) before the assembly of the L-ring? This will be helpful as many people would probably only look at the figures and not read the full text, therefore, indicating this in the figure would be useful.

3- The Jensen lab, Beeby lab, Liu lab and Bai lab recently published papers about the presence of PL-subcomplexes as a relic structure after the disassembly of the motor and these PL-

subcomplexes are plugged to prevent the formation of a hole in the outer membrane (see Refs. (2-7)). As this manuscript describes the first high-resolution structure of the PL-rings, I would encourage the authors to add a paragraph about the presence of relic plugged PL-subcomplexes and cite the relevant papers (Ref. (2-7)). Does their structure help to understand how these rings are present stably in the outer membrane in the absence of the rod? I think the fact that PL-rings can present stably, independent of the rest of the motor is very related to this work and should be mentioned. This paragraph and the relevant literature might be added in the introduction (for example line 58 where the authors discuss the stability of PL rings) or somewhere in the discussion.

4- Line 107, the authors say that there is a blurred density beneath the P-ring (FlgI), which is likely a part of FlgI. Can the authors speculate which part of FlgI this might be based on their structure?

5- Lines 191-196, the authors suggest that PL-rings are required for the assembly of FlgD (the hook capping protein) and FlgE (the hook protein) because these proteins cannot be detected in the culture (which means they have not been secreted through the outer membrane) when the PL-rings are not present. I think this is just related simply to the fact that the L-ring protein makes a hole in the outer membrane when it assembles (see Ref. (3)), and so when it is not there, FlgD and FlgE cannot be secreted outside the cell because there is no hole. This is also supported by the work of Kubori et al. (See Ref. (8)) which showed that sometimes the hook protein (FlgE) can even assemble before the P-ring protein (FlgI), see Figures 9 and 10 in reference (8), and note that Figure 9 is for flgI mutant indicating that the observed structures are assembly and not disassembly ones. As it is written now, that part (lines 191-196) is confusing by implying a real role for the PL-rings in the assembly of FlgD and FlgE which I do not think is correct. I would recommend paraphrasing these few lines in light of a hole formation by FlgH.

6- I liked how the authors compared the structure of PL-rings to that of other secretion systems, as detailed in Extended Data Figures 6 and 7. However, I found the comparison rather descriptive and I missed clear sentences of what we learn from them. For example, despite the structural similarity between the bacterial flagellum and the injectisome, it is known that the PL-rings, which are an ancient component of the flagellum (see Ref. (5)), are specific to the flagellum and share low similarity to the secretin of the injectisome (see Ref. (9)). Now that we have the high-resolution structure of PL-rings and we can compare them to other secretion systems, what do we learn about how these systems might have evolved? Does this comparison between the high-resolution structures of these components of various secretion systems teach us something new about their evolution and any evolutionary links between them? Can this structure of PL-rings provide an insight into why they are specific to the flagellum and different from the secretin of the injectisome? I think it would be interesting if the authors could write few clear sentences and thoughts in that direction based on the structural comparison they have already performed.

Minor points:

1- Line 65 (...the LP ring assemble...), please add "s" to the word assemble as the word "LP ring" is used in the singular form, or preferably use "LP rings" in the plural form as these are indeed two separate rings.

2- Line 139 (...protein of type VIa pili,...), please note that it should be "type IVa" and not "VIa". In other words, it is type four and not type six pili. The same mistake is done in Extended Data Fig. 6 (both in the figure legend and in panel c). Also, the word "type" is missed from the legend of Extended Data Fig. 6 and in panel c from that figure.

3- The citation of Extended Data figures in the main text is not done in a sequential manner (For example, Extended Data Figure 9 is cited in the text before Extended Data Figure 8). Probably just flip the sequence of these figures in the SI, that would be easier than changing the text?

4- Throughout the text, FlgI three domains are referred to as: FlgI-IRU (upper inner ring), FlgI-IRL (lower inner ring) and FlgI-OR (outer ring). Then the two conserved hydrophobic pockets in FlgI-

IRU are referred to as IR (inner ring) and OR (outer ring) pockets. This can be confusing to the reader. Probably just refer to the conserved pockets as the inner pocket (IP) and outer pocket (OP) to avoid using IR and OR abbreviations which are already used for the major domains of this protein?

5- Can the authors write the cryoEM single particle image analysis (lines 94-104) a bit clearer? Extended Data Figure 2 is clear but the description part (lines 94-104) is a bit ambiguous.

6- Labels of adjacent FlgH subunits in Fig. 2 b are too small to be seen. Also, it is really difficult to read Extended Data Figures 4, 5 and 7 (panel d). Just as an example, Extended Data Figure 4 b, it is difficult to read the axes of that figure. But many other parts of these three figures are too small to be read.

7- In Figure 3 b and c, what does the letter (V) before (WT) refer to? Probably to empty expression vector? Please just mention explicitly in the legend what it means?

8- In Extended Figure 8 panel e, the word "rod" at the center of that panel is difficult to see. Please make it either clearer or put it outside the figure and use an arrow?

9- Line 263 (So, these two lysines....), I would use another word instead of "so", like "Hence" or "Therefore".

Good luck!

References:

1. Johnson S, Furlong EJ, Deme JC, Nord AL, Caesar J, Chevance FFV, Berry RM, Hughes KT, Lea SM. 2020. Molecular structure of the intact bacterial flagellar basal body. preprint, Microbiology.
2. Ferreira JL, Gao FZ, Rossmann FM, Nans A, Brenzinger S, Hosseini R, Wilson A, Briegel A, Thormann KM, Rosenthal PB, Beeby M. 2019. γ -proteobacteria eject their polar flagella under nutrient depletion, retaining flagellar motor relic structures. PLOS Biology 17:e3000165.
3. Kaplan M, Subramanian P, Ghosal D, Oikonomou CM, Pirbadian S, Starwalt-Lee R, Mageswaran SK, Ortega DR, Gralnick JA, El-Naggar MY, Jensen GJ. 2019. In situ imaging of the bacterial flagellar motor disassembly and assembly processes. The EMBO Journal e100957.
4. Zhu S, Schniederberend M, Zhitnitsky D, Jain R, Galán JE, Kazmierczak BI, Liu J. 2019. In Situ Structures of Polar and Lateral Flagella Revealed by Cryo-Electron Tomography. Journal of Bacteriology 201.
5. Kaplan M, Sweredoski MJ, Rodrigues JPGLM, Tocheva EI, Chang Y-W, Ortega DR, Beeby M, Jensen GJ. 2020. Bacterial flagellar motor PL-ring disassembly subcomplexes are widespread and ancient. Proceedings of the National Academy of Sciences 201916935.
6. Zhuang X, Guo S, Li Z, Zhao Z, Kojima S, Homma M, Wang P, Lo C, Bai F. 2020. Live-cell fluorescence imaging reveals dynamic production and loss of bacterial flagella. Molecular Microbiology 114:279–291.
7. Zhu S, Gao B. 2020. Bacterial Flagella Loss under Starvation. Trends in Microbiology <https://doi.org/10.1016/j.tim.2020.05.002>.
8. Kubori T, Shimamoto N, Yamaguchi S, Namba K, Aizawa S-I. 1992. Morphological pathway of flagellar assembly in *Salmonella typhimurium*. Journal of Molecular Biology 226:433–446.
9. Diepold A, Armitage JP. 2015. Type III secretion systems: the bacterial flagellum and the injectisome. Philosophical Transactions of the Royal Society B: Biological Sciences 370:20150020.

Reviewer #3 (Remarks to the Author):

The authors report the structure of the flagellar LP ring (bushing) by electron cryomicroscopy. They reveal 26-fold rotational symmetry, intersubunit interactions and charge patterns. Based on

these findings, they identify a number of amino acids which may be important for assembly and function of the LP ring, and test a number of these using motility, stability and secretion assays. The study is generally convincing and will be of interest to the field and more widely. However, to strengthen the conclusions, there are a number of points which need addressing:

1. Some parts of the manuscript were difficult to follow, in particular with respect to protein names and location within the machinery. A simple schematic overview would help considerably.
2. Line 93 - a sentence is missing at the beginning of the results section to shortly describe how the protein sample was made - it jumps straight into imaging and gives the impression that the sample was provided.
3. Line 104 - explain more clearly why another 3D image of the HBB was needed with C1 symmetry, i.e. why couldn't the one already determined be used.
4. Line 113 - point out how much of FlgH residues 1-211 correspond to (e.g. in %) and in line 129 the same for FlgI.
5. Line 138 - "type VIa" should be "type IVa". Also spotted in the legend to Ext Data Fig. 6. Also, why was the *M. xanthus* type IVa structure chosen for comparison? It is based on homology modelling and subsequent docking into a 3-4 nm resolution subtomogram average, whereas the PilQ secretin from *V. cholerae* was built directly from a 2.7Å cryoEM map (<https://www.nature.com/articles/s41467-020-18866-y>). How does the *Vibrio* structure compare? Whilst it is clear why comparisons are made to secretin proteins in general (line 123), the reason for comparison to MotY is less obvious. It would be helpful to point out why these proteins were chosen specifically for comparison and what this means.
6. There is considerable discussion around proximity of charged residues to the rod e.g. starting at line 146, 168 and many other instances. This reviewer found this argument to be confusing until it was noted that the relevant data are shown in Fig. 4a-e, but these figures are not referred to anywhere. References need to be included throughout, in particular at first mention on line 146.
7. Line 156 - "hydrophobic interactions.. Fig. 2e". It is not easy to see that these are hydrophobic from the figure.
8. Line 180 - spell out the aim of the experiment in words e.g. mutations to reverse/neutralise charge rather than the reader needing to read lines of mutant names to figure out the point of the experiment. Also explain the results in similar wording rather than just listing mutant names.
9. Line 186 - 189 needs further clarification/rewording. It is stated that two mutants were impaired in FlgI protein stability, explaining a no motility phenotype, which is shown in the blot. However, there are other mutants that also show a no motility phenotype. These show wt levels of FlgI, but show a reduction in the amount of FlgE (hook) and FliC (filament). Presumably the no motility phenotype here is due to there being less assembled flagella? A short explanation would be helpful.
10. Line 199 - comparisons are made between levels of secreted proteins and swarm motility in Fig. 3b/c, but these data do not appear to have been quantified (neither motility assays nor blots). How was the 2-fold reduction deduced? To make such statements (also with respect to the "lower" level of secretion by K95D), quantifications in triplicate should be carried out, with appropriate statistics performed. Also, these data do not demonstrate anything about kinetics and rates of assembly, so should be reworded.
11. Lines 180 - 206 - considerable weight is placed on the roles of the two lysine residues in P ring assembly. To explain the results of the motility assays and to provide stronger links with assembly/function, the number of filaments on the surface of cells should be assessed directly, either by electron or fluorescence microscopy.
12. Line 233 - MotA is mentioned for the first time and was not in the introduction. See point 1.
13. Line 254 - It is not clear why the structure of FlgG is being compared to FlgE (Ext Data 8) and therefore what the discussion around this point means.
14. Line 269 - the manuscript ends very abruptly. The authors should provide a couple of sentences about the wider context of their findings.
15. Line 380 & 387 - The "WT" cells used in motility and expression/secretion assays are not actually wild-type. This should be made clear in the figure/legend. The authors should clarify if this mutant behaves in the same way as wild-type.

Extra comments relating to figures

16. Sky blue and cyan look quite similar. Suggest using more contrasting colours for the L and P

rings in Fig. 1, Fig. 4, Ext. Fig. 3 etc.

17. Fig. 1c typo – middle layer; and “beige” does not look beige

18. Show how FlgH and FlgI in Fig. 1 panel d relate to the figure in panel c (e.g. by including another panel showing the two proteins in just two colours). It is not entirely clear if FlgH forms the L ring and FlgI the P ring exclusively, or if parts of both proteins contribute to both rings. This could also be made clearer in the text in Introduction and Results.

19. Fig. 3a – it would be helpful to label the distal rod and the inner surface of the LP ring. Also, the negatively charged belts are not very convincing on the left, patches may be a better word.

20. Fig. 3a - The Lys-63 and Lys-95 (line 175) don't actually seem to sit within the blue positive charge belt. Are they correctly positioned in the figure, or are there other positively charged residues that sit above them?

21. Fig. 4a – multiple elements are labelled in grey, it would be better to change some colours

22. How were the sequence conservation structure plots made e.g. in Ext. Fig. 4c/5b/c/9?

23. Ext. Data Fig. 6 – it would be better to label the organism/protein at the top instead of part way down

24. Ext. Data Fig. 9 – only FlgI is labelled, add label for FlgH for clarity

Response to the reviewers' comments:

To Reviewer #1:

This is an elegant structural analysis of the LP ring of the flagellar motor. This cryo-EM revolution is great, it might be time at last to really understand the flagellar motor. Within this structure, the flagellar driveshaft rotates at speeds of hundreds to a couple of thousand cycles per second. It's truly mind-boggling. The detailed analysis of inter-subunit interactions helps to explain the incredible stability of this bushing and resistance to harsh chemical treatments. The structure also solves a prior mystery regarding differences in polyrod alleles of *flgG* that either do or do not (G53R & G183R) allow P-ring formation. I wonder if K63G and K95G substitutions in FlgI could suppress this (I am not requesting this just thinking about the possibility). One important dilemma is the fact that the P-ring must initially form around the FlgG distal rod structure, but the separate from the rod upon completion of the PL-ring structure. It's a beautiful structure.

Re: Thank you so much for your supportive comments. At the moment, we do not know whether substitutions of K63G and K95G in FlgI suppress the poly-P ring formation phenotype by G53R and G183R mutation in FlgG. But, to address to this interesting question, we will carry out mutational analysis in the future.

To Reviewer #2: Mohammed Kaplan

This nice manuscript by (Tomoko Yamaguchi, Fumiaki Makino et al.) presents the first structure of the P- and L-rings of the flagellar motor at a resolution of 3.5 Å using single particle reconstruction cryo-EM. This structure allowed the authors to present a model of how these two rings might assemble and function as a bushing during the motor rotation. In recent months, high-resolution structures of various parts of the motor have been published, so this work is timely and helpful for the field. The manuscript (as is usually the case with the work done by Prof. Namba and colleagues) is a solid piece of work that deserves to be published in a prestigious journal like Nature Communications.

Re: Thank you for your kind note and helpful comments for revising the manuscript.

I have the following suggestions to the authors:

1- Recently, a manuscript from the lab of Prof. Susan Lea at Oxford (Johnson et al., see ref. (1)) was posted on the bioRxiv reporting the structure of the flagellar PL-rings (also from Salmonella) with other parts of the basal body like the rod and MS-ring. However, in their structure they saw that there is an extra density surrounding the FlgH density and after doing proteomics they identified it as YecR. It is interesting that Yamaguchi and colleagues do not see an extra density around FlgH in their structure. Do they have an explanation for this?

Re: Yes, it is an interesting difference between the two structures. This is probably due to the difference in the preparation methods. We adjusted the pH to 10.5 after cell lysis while Johnson et al did not do such alkaline pH treatment. The higher pH treatment may have caused YecR dissociation.

Johnson et al. hypothesize that YecR might remodel the lipids to help the assembly of FlgH (L-ring) because the lipid bilayer surrounding FlgH is shorter than usual in their structure. Can the authors here check the lipid bilayer thickness around FlgH in their structure? Is it also thinner? Probably the YecR ring disassembles at a certain stage after FlgH assembly? or do the authors have another explanation (hypothesis) for its absence from their structure? I think adding something (could be few sentences) regarding this point would be helpful for the reader.

Re: We also observed some extra densities, possibly representing polar groups of the detergent in the top portion of the LP ring, as shown in the figure attached below. The distance between

them is about 2.5 nm, which is smaller than that of lipid bilayers (about 4.5 nm). However, in the absence of lipid bilayer in the detergent-purified basal body, it is difficult to tell how the lipids surround FlgH of the L ring. The fact that Johnson et al observed YecR in their basal body structure means that YecR is associated with the L ring in the *in situ* structure.

2- The authors indicate that the P-ring (which assembles before the L-ring) undergoes a conformational change after the assembly of the L-ring around the rod (lines 268-272). Do they also think, based on their structure, that the P-ring is located closer to the MS-ring before the L-ring assembles as suggested by Johnson et al.? probably the authors can check for that if they already have a subset of particles that only have the P-ring? If not, the authors can probably indicate in Fig. 4 f that somehow the P-ring is different (in whatever way the authors find suitable) before the assembly of the L-ring? This will be helpful as many people would probably only look at the figures and not read the full text, therefore, indicating this in the figure would be useful.

Re: We compared the distances between the S ring and P ring for the basal bodies with and without the L ring. We purified the basal bodies from two mutant strains, one forming the L ring and the other not, observed them by negative stain EM (nsEM) to measure the distances and compared them with that measured on our cryoEM structure. The S-P ring distances measured on the 2D class average nsEM images from about 100 LP-ring basal bodies and about 90 P-ring basal bodies were 12.3 nm and 11.6 nm, respectively, and that of our cryoEM structure was 12.5 nm. So, our data agrees with what Johnson et al described; the P-ring is located closer to the MS-ring before L-ring formation.

3- The Jensen lab, Beeby lab, Liu lab and Bai lab recently published papers about the presence of PL-subcomplexes as a relic structure after the disassembly of the motor and these PL-subcomplexes are plugged to prevent the formation of a hole in the outer membrane (see Refs. (2-7)). As this manuscript describes the first high-resolution structure of the PL-rings, I would encourage the authors to add a paragraph about the presence of relic plugged PL-subcomplexes and cite the relevant papers (Ref. (2-7)). Does their structure help to understand how these rings are present stably in the outer membrane in the absence of the rod? I think the fact that PL-rings can present stably, independent of the rest of the motor is very related to this work and should be mentioned. This paragraph and the relevant literature might be added in the introduction (for

example line 58 where the authors discuss the stability of PL rings) or somewhere in the discussion.

Re: Thank you for your suggestion. We cited the suggested papers to describe the mechanically stable property of the LP ring in the text.

4- Line 107, the authors say that there is a blurred density beneath the P-ring (FlgI), which is likely a part of FlgI. Can the authors speculate which part of FlgI this might be based on their structure?

Re: Based on our structural model of FlgI, the blurred density is likely correspond to missing residues 264-295. We modified the sentence as follows.

“FlgI-IR_L is formed by the C-terminal chain of FlgI and contains a highly flexible loop (residues 264-295) that is likely to form the extra ring density beneath the P ring (Fig. 2b).”

5- Lines 191-196, the authors suggest that PL-rings are required for the assembly of FlgD (the hook capping protein) and FlgE (the hook protein) because these proteins cannot be detected in the culture (which means they have not been secreted through the outer membrane) when the PL-rings are not present. I think this is just related simply to the fact that the L-ring protein makes a hole in the outer membrane when it assembles (see Ref. (3)), and so when it is not there, FlgD and FlgE cannot be secreted outside the cell because there is no hole. This is also supported by the work of Kubori et al. (See Ref. (8)) which showed that sometimes the hook protein (FlgE) can even assemble before the P-ring protein (FlgI), see Figures 9 and 10 in reference (8), and note that Figure 9 is for flgI mutant indicating that the observed structures are assembly and not disassembly ones. As it is written now, that part (lines 191-196) is confusing by implying a real role for the PL-rings in the assembly of FlgD and FlgE which I do not think is correct. I would recommend paraphrasing these few lines in light of a hole formation by FlgH.

Re: You are correct that the LP ring complex is not required the assembly of FlgD and FlgE at the rod tip. During the transition state from the completion of rod assembly to the initiation of hook assembly, the L ring forms a pore in the outer membrane so that the hook can elongate into the cell exterior. So, we modified the sentence to “... the LP ring is required for forming the pore in the outer membrane to expose the distal end of the rod in the cell exterior to allow hook assembly outside the cell body.”

6- I liked how the authors compared the structure of PL-rings to that of other secretion systems, as detailed in Extended Data Figures 6 and 7. However, I found the comparison rather descriptive and I missed clear sentences of what we learn from them. For example, despite the structural similarity between the bacterial flagellum and the injectisome, it is known that the PL-rings, which are an ancient component of the flagellum (see Ref. (5)), are specific to the flagellum and share low similarity to the secretin of the injectisome (see Ref. (9)). Now that we have the high-resolution structure of PL-rings and we can compare them to other secretion systems, what do we learn about how these systems might have evolved? Does this comparison between the high-resolution structures of these components of various secretion systems teach us something new about their evolution and any evolutionary links between them? Can this structure of PL-rings provide an insight into why they are specific to the flagellum and different from the secretin of the injectisome? I think it would be interesting if the authors could write few clear sentences and thoughts in that direction based on the structural comparison they have already performed.

Re: Thank you for your suggestions. We added another paragraph comparing the LP ring with other secretion systems in the last part of this section to discuss this. As you mentioned, FlgH and FlgI forming the LP ring are specific to the flagellum, and there are no homologs in the injectisome, but the L ring formed by FlgH has a cylindrical β barrel structure similar to those of the outer membrane secretin of the injectisome. Because the LP ring functions as a secretin-like pore-forming annulus to allow the transport of FlgD and FlgE into the cell exterior in addition to the molecular bushing function, this is probably the result of convergent evolution from different origins.

Minor points:

1- Line 65 (...the LP ring assemble....), please add “s” to the word assemble as the word “LP ring” is used in the singular form, or preferably use “LP rings” in the plural form as these are indeed two separate rings.

Re: Corrected to “assembles”.

2- Line 139 (...protein of type VIa pili,...), please note that it should be “type IVa” and not “VIa”. In other words, it is type four and not type six pili...

Re: Corrected.

Also, the word “type” is missed from the legend of Extended Data Fig. 6 and in panel c from that figure.

Re: Corrected.

3- The citation of Extended Data figures in the main text is not done in a sequential manner....

Re: Corrected.

4- Throughout the text, FlgI three domains are referred to as: FlgI-IRU (upper inner ring), FlgI-IRL (lower inner ring) and FlgI-OR (outer ring). Then the two conserved hydrophobic pockets in FlgI-IRU are referred to as IR (inner ring) and OR (outer ring) pockets. This can be confusing to the reader. Probably just refer to the conserved pockets as the inner pocket (IP) and outer pocket (OP) to avoid using IR and OR abbreviations which are already used for the major domains of this protein?

Re: We agree that OR-pocket and IR-pocket may be confusing to the reader. So, we replaced them to “inner pocket” and “outer pocket”, respectively.

5- Can the authors write the cryoEM single particle image analysis (lines 94-104) a bit clearer? Extended Data Figure 2 is clear but the description part (lines 94-104) is a bit ambiguous.

Re: We added a few sentences to make the process of image analysis clearer.

6- Labels of adjacent FlgH subunits in Fig. 2 b are too small to be seen. Also, it is really difficult to read Extended Data Figures 4, 5 and 7 (panel d). Just as an example, Extended Data Figure 4 b, it is difficult to read the axes of that figure. But many other parts of these three figures are too small to be read.

Re: We made the labels larger.

7- In Figure 3 b and c, what does the letter (V) before (WT) refer to? Probably to empty expression vector? Please just mention explicitly in the legend what it means?

Re: We explained it in the text and legend .

8- In Extended Figure 8 panel e, the word “rod” at the center of that panel is difficult to see. Please make it either clearer or put it outside the figure and use an arrow?

Re: We modified it as suggested.

9- Line 263 (So, these two lysines....), I would use another word instead of “so”, like “Hence” or “Therefore”.

Re: Changed to “Hence”.

Good luck!

Re: Thank you.

To Reviewer #3

The authors report the structure of the flagellar LP ring (bushing) by electron cryomicroscopy. They reveal 26-fold rotational symmetry, intersubunit interactions and charge patterns. Based on these findings, they identify a number of amino acids which may be important for assembly and function of the LP ring, and test a number of these using motility, stability and secretion assays. The study is generally convincing and will be of interest to the field and more widely. However, to strengthen the conclusions, there are a number of points which need addressing:

Re: Thank you so much for your kind and helpful comments for revising the manuscript.

1. Some parts of the manuscript were difficult to follow, in particular with respect to protein names and location within the machinery. A simple schematic overview would help considerably.

Re: We provided a schematic diagram of the flagellum as Fig. 1 to explain the location of its structural subunits.

2. Line 93 - a sentence is missing at the beginning of the results section to shortly describe how the protein sample was made – it jumps straight into imaging and gives the impression that the sample was provided.

Re: We modified the first sentence as follows.

“We purified the hook-basal body (HBB) complex from the *Salmonella* HK1002 cells (see Methods), collected cryoEM images, analyzed them by single particle image analysis using RELION²⁵, and analyzed the LP ring structure in the HBB.”

3. Line 104 – explain more clearly why another 3D image of the HBB was needed with C1 symmetry, i.e. why couldn't the one already determined be used.

Re: Because we needed to improve the quality and resolution of the HBB map for precise determination of the relative positions of the rod and LP ring. We made it clearer by modifying the relevant sentences as follows.

“To determine the relative positioning of the rod and LP ring precisely, a 3D image of the HBB was again reconstructed with C1 symmetry from 14,370 HBB images extracted with a larger box size from the same data sets, with the 3.5 Å resolution map of the LP ring used as a reference for the refinement, and this produced a 6.9 Å resolution density map (EMD-30409) with a better quality than the initial HBB map. Although the global resolution of this map was the same as that of the initial one, the local resolution was improved (Fig. 2a, Supplementary Fig. 2a; see Methods for more detail)”

4. Line 113 – point out how much of FlgH residues 1-211 correspond to (e.g. in %) and in line 129 the same for FlgI.

Re: We stated them in the text. They are 100% of FlgH and 87% of FlgI.

5. Line 138 – “type VIa” should be “type IVa”. Also spotted in the legend to Ext Data Fig. 6.

Re: Corrected.

Also, why was the *M. xanthus* type IVa structure chosen for comparison? It is based on homology modelling and subsequent docking into a 3-4 nm resolution subtomogram average, whereas the **PilQ secretin from *V. cholerae* was built directly from a 2.7Å cryoEM map** (<https://www.nature.com/articles/s41467-020-18866-y>). How does the *Vibrio* structure compare?

Re: We did not have access to the structure from the 2.7 Å cryoEM map when we prepared the manuscript. However, now it is now available, we replaced the *M. xanthus* type IVa structure with the *V. cholerae* PilQ secretin structure.

Whilst it is clear why comparisons are made to secretin proteins in general (line 123), the reason for comparison to MotY is less obvious. It would be helpful to point out why these proteins were chosen specifically for comparison and what this means.

Re: The FlgI-OR domain is located on the outermost surface of the P ring and is likely to interact with the peptidoglycan (PG) layer. In order to give more supporting evidence that FlgI-OR associates with the PG layer, we compared it with other secretion proteins and showed that they are located in the almost same position in the PG layer. The comparison with MotY was just to confirm this because MotY also has a PG-binding (PGB) domain (Kojima, S. *et al.* PNAS 2008), and we actually found a significant structural similarity between FlgI-OR and the PGB domain of MotY. We added a paragraph in the last part of this section to explain it.

6. There is considerable discussion around proximity of charged residues to the rod e.g. starting at line 146, 168 and many other instances. This reviewer found this argument to be confusing until it was noted that the relevant data are shown in Fig. 4a-e, but these figures are not referred to anywhere. References need to be included throughout, in particular at first mention on line 146.

Re: We now show the charged residues (Lys-63 and Lys-95) of FlgI in Fig. 2a right panel and refer to them in the text.

7. Line 156 – “hydrophobic interactions. Fig. 2e”. It is not easy to see that these are hydrophobic from the figure.

Re: We believe it is not so difficult to see the hydrophobic interactions because all the hydrophobic side chains are depicted in sticks.

8. Line 180 – spell out the aim of the experiment in words e.g. mutations to reverse/neutralise charge rather than the reader needing to read lines of mutant names to figure out the point of the experiment. Also explain the results in similar wording rather than just listing mutant names.

Re: We added the following sentence to clarify why reverse/neutralize charge mutations were examined. We also added explanations by wording to make the results easier to read.

“To examine whether these positive charges contribute to the P ring assembly, we replaced Lys-63 and Lys-95 with alanine or oppositely charged residue (Asp), constructed eight *flgI* mutants, *flgI(K63A)*, *flgI(K63D)*,”.

9. Line 186 - 189 needs further clarification/rewording. It is stated that two mutants were impaired in FlgI protein stability, explaining a no motility phenotype, which is shown in the blot. However, there are other mutants that also show a no motility phenotype. These show wt levels of FlgE (hook) and FliC (filament). Presumably the no motility phenotype here is due to there being less assembled flagella? A short explanation would be helpful.

Re: You are right. The non-motile phenotype of *flgI* mutants is due to a defect in P ring assembly. We added the following sentence to explain the relationship between the non-motile phenotype and the assembly of flagellum and added another figure as Supplementary Fig. 9a.

“The flagellum is necessary for motility, and the disturbance of P ring assembly leads to the inhibition of flagellum formation and no motility. Since the swarm size and the number of the filaments were correlated (Supplementary Fig. 9b), these results indicate that the positive charges of Lys-63 and Lys-95 are both critical for FlgI to form the P ring.”

10. Line 199 - comparisons are made between levels of secreted proteins and swarm motility in Fig. 3b/c, but these data do not appear to have been quantified (neither motility assays nor blots). How was the 2-fold reduction deduced? To make such statements (also with respect to the “lower” level of secretion by K95D), quantifications in triplicate should be carried out, with appropriate statistics performed. Also, these data do not demonstrate anything about kinetics and rates of assembly, so should be reworded.

Re: Thank you for your suggestions. We quantified the diameter of motility rings and the amount of secreted FlgE and FliC proteins and performed their statistical analysis (Fig. 5 and Supplementary Fig. 9). We also rephrased “reduced the rate of P ring assembly ...” to “inhibited P ring assembly ...”.

11. Lines 180 – 206 - considerable weight is placed on the roles of the two lysine residues in P ring assembly. To explain the results of the motility assays and to provide stronger links with assembly/function, the number of filaments on the surface of cells should be assessed directly, either by electron or fluorescence microscopy.

Re: Thank you for your advice. We measured the number of filaments by electron microscopy of cells negatively stained with 2 % uranyl acetate and confirmed that the number of filaments and the swarm size were correlated. We added such data as Supplementary Fig. 9b and explained it in the text.

12. Line 233 - MotA is mentioned for the first time and was not in the introduction. See point 1.

Re: We introduced MotA in Introduction and labeled it in the schematic diagram in Fig. 1.

13. Line 254 – It is not clear why the structure of FlgG is being compared to FlgE (Ext Data 8) and therefore what the discussion around this point means.

Re: FlgG forms the rigid and straight distal rod, while FlgE forms the curved and flexible hook. Even with such distinct differences in the physical and mechanical properties of the rod and hook, the sequences and structures of FlgG and FlgE molecules are very similar to each other for their two common domains of D0 and D1. The only difference is a long “L-stretch” loop of FlgG that is not present in FlgE and makes the rod rigid and straight, and all the polyrod mutations we mentioned in this paper are localized in this L-stretch. This is the reason why we refer to FlgE in this sentence.

14. Line 269 - the manuscript ends very abruptly. The authors should provide a couple of sentences about the wider context of their findings.

Re: We added a few sentences as follows.

“Understanding the mechanical and dynamic properties of the LP ring as a nanoscale bushing would also be quite interesting in physics and useful in nanotechnology applications, but it requires fully atomic molecular dynamics simulations of the rod rotation within the LP ring. Although it would need to deal with tens of millions of atoms and therefore is not a simple task, such study is also underway because it is now feasible by the development of high-speed computers, such as Fugaku.”

15. Line 380 & 387 – The “WT” cells used in motility and expression/secretion assays are not actually wild-type. This should be made clear in the figure/legend. The authors should clarify if

this mutant behaves in the same way as wild-type.

Re: We clarified what the “WT” and the “V” refers to by following sentence.

“We used a plasmid vector (pET22b, V) and a wild-type plasmid [pTY03 (Supplementary Table 1), WT] as negative and positive controls, respectively.”

Extra comments relating to figures

16. Sky blue and cyan look quite similar. Suggest using more contrasting colours for the L and P rings in Fig. 1, Fig. 4, Ext. Fig. 3 etc.

Re: To make the difference clearer, we changed the color of the L ring from sky blue to purple.

17. Fig. 1c typo – middle layer; and “beige” does not look beige

Re: We corrected Fig. 1c typo and rephrased “beige” as “light yellow”

18. Show how FlgH and FlgI in Fig. 1 panel d relate to the figure in panel c (e.g. by including another panel showing the two proteins in just two colours). It is not entirely clear if FlgH forms the L ring and FlgI the P ring exclusively, or if parts of both proteins contribute to both rings. This could also be made clearer in the text in Introduction and Results.

Re: We modified Fig. 2 (Former Fig. 1) and changed the color of FlgH forming the L ring to make it easier to see that FlgH and FlgI form the L and P ring more or less exclusively. We also added the following sentence in the text.

“The FlgH forms the L ring and FlgI forms the P ring more or less exclusively (Fig. 2).”

19. Fig. 3a – it would be helpful to label the distal rod and the inner surface of the LP ring. Also, the negatively charged belts are not very convincing on the left, patches may be a better word.

Re: We added labels “Distal rod” and “Inner surface of the LP ring” under the panels. We replaced the word “belts” to “patches” as your advice.

20. Fig. 3a - The Lys-63 and Lys-95 (line 175) don't actually seem to sit within the blue positive charge belt. Are they correctly positioned in the figure, or are there other positively charged residues that sit above them?

Re: Yes, the positions of Lys-63 and Lys-95 are correct. This is a Coulomb potential map, and the positive belt is not only formed by Lys-63 and Lys-95 but also by the amide groups of Gln-32, Gln-35, and Asn-93 above them.

21. Fig. 4a – multiple elements are labelled in grey, it would be better to change some colours

Re: That is simply to highlight the LP ring.

22. How were the sequence conservation structure plots made e.g. in Ext. Fig. 4c/5b/c/9?

Re: We used UCSF Chimera. We added explanations in the legends of Fig. 4 and Supplementary Figs. 4, 5, and 8.

23. Ext. Data Fig. 6 – it would be better to label the organism/protein at the top instead of part way down

Re: We placed the labels of proteins and organisms at the top of each panel.

24. Ext. Data Fig. 9 – only FlgI is labelled, add label for FlgH for clarity

Re: We added labels for FigH.

Thank you again for your detailed comments for improvement of the manuscript.

REVIEWERS' COMMENTS

Reviewer #2 (Remarks to the Author):

Report for "Structure of the molecular bushing of the bacterial flagellar motor"

Reviewer: Mohammed Kaplan

The authors have addressed all my concerns adequately in their revised manuscript. Congratulations!

I have the following minor suggestions and questions to the authors:

1- Isn't it strange that the secretion of FliC and FlgE in flgI(K63A) and flgI(K95A) are at the wild-type level but the number of filaments and the swarm motility assay are significantly lower than the wild type? Probably the immunoblotting is not sensitive enough at certain levels (in other words, maybe it gets saturated at certain levels)?

2- Figure 5 panel (a): why are there two panels for the motility assays of the (V) and the (k63D) and one panel for all the other mutants?

3- Line 233: "swarm" not "swam". Also, the same mistake in the rebuttal file (response to reviewers).

4- The name of the journal of reference 5 is omitted (it is Molecular Microbiology).

5- Lines 280 and 297: probably remove the abbreviation (Ref.)? all the other references are cited just as numbers except in these two lines they are cited as (Ref. number).

6- Legend of figure 1 (line 673): "proximal" not "proxima"

7- Legend of figure 1 (line 675): two spaces are left between OM and the dot before it.

8- Legend of figure 1 (line 676): "peptidoglycan" not "peptide glycan".

9- Legend of figure 1 (line 676): "inner membrane" not "inter membrane".

10- Legend of Figure 2 (line 683): "b, The...." not "b, the....." (to be consistent with the rest of the manuscript).

11- Legend of Figure 4 (line 719): "calculated and visualized by Chimera" not "calculate and visualized by Chimera".

12- Figure 5: the label "a" is cropped.

13- Figure 6a: I really struggled to see the grey polar residues.

14- The title of the SI file is inconsistent with the title of the main article. It is missing the word "bacterial".

15- Figure S1: the enlargement in panel (a) is covering the letter "i" in the word "ring". Also, use "MS & C rings" in the plural form as these are two different rings

16- There are multiple typos in Figure S2: "polar coordinates" not "polar coordinate", "After" not "Afer", "LP ring &" not "LP ring&", "Ignore" not "Ignor". Please check this figure carefully.

17- Figure S7 panel (b): in this table "RMSD" is misspelled.

18- Figure S9 panel (b): add (N.F.) to the column of V? I presume you did not see filaments here as FliC and FlgE were not secreted to the culture?

19- Figure S10 d: "predicted $\Delta 54-57$ " not "predected $\Delta 54-57$ ".

Reviewer #3 (Remarks to the Author):

The authors have done a very good job of revising the manuscript. All of my comments have been addressed comprehensively.

Response to the reviewers' comments:

To Reviewer #2:

The authors have addressed all my concerns adequately in their revised manuscript. Congratulations!

Thank you for your helpful comments and questions. We really appreciate it.

I have the following minor suggestions and questions to the authors:

We also appreciate your careful review of the manuscript and further comments for revision.

1- Isn't it strange that the secretion of FliC and FlgE in flgI(K63A) and flgI(K95A) are at the wild-type level but the number of filaments and the swarm motility assay are significantly lower than the wild type? Probably the immunoblotting is not sensitive enough at certain levels (in other words, maybe it gets saturated at certain levels)?

Re: We think the immunoblots are sensitive enough to reflect the secretion levels of FliC and FlgE. But because other flagellar proteins, such as FlgG, FlgK, FlgL and FlgD, also need to be exported for assembly of the rod, hook and filament, and their secretion levels may somehow be reduced by these flgI mutations, it is difficult to say what is actually the cause of the reduced filament number and motility.

2- Figure 5 panel (a): why are there two panels for the motility assays of the (V) and the (k63D) and one panel for all the other mutants?

Re: That is to make it easier to compare the motility of mutants with negative and positive controls (V or WT). In the top panel, the mutants are compared with WT to show how much their motility is decreased. In the middle panel, the mutants are compared with V to show their motility. In the bottom panel, the mutants are compared with V to show that they are not motile (within this incubation time).

3- Line 233: "swarm" not "swam". Also, the same mistake in the rebuttal file (response to reviewers).

Re: Corrected.

4- The name of the journal of reference 5 is omitted (it is Molecular Microbiology).

Re: Corrected.

5- Lines 280 and 297: probably remove the abbreviation (Ref.)? all the other references are cited just as numbers except in these two lines they are cited as (Ref. number).

Re: It is the format of this journal that the reference number after numerical characters should be cited as such.

6- Legend of figure 1 (line 673): "proximal" not "proxima"

Re: Corrected.

7- Legend of figure 1 (line 675): two spaces are left between OM and the dot before it.

Re: Corrected.

8- Legend of figure 1 (line 676): “peptidoglycan” not “peptide glycan”.

Re: Corrected.

9- Legend of figure 1 (line 676): “inner membrane” not “inter membrane”.

Re: Corrected.

10- Legend of Figure 2 (line 683): “b, The...” not “b, the....” (to be consistent with the rest of the manuscript).

Re: Corrected.

11- Legend of Figure 4 (line 719): “calculated and visualized by Chimera” not “calculate and visualized by Chimera”.

Re: Corrected.

12- Figure 5: the label “a” is cropped.

Re: Corrected.

13- Figure 6a: I really struggled to see the grey polar residues.

Re: We changed it to pink.

14- The title of the SI file is inconsistent with the title of the main article. It is missing the word “bacterial”.

Re: Corrected.

15- Figure S1: the enlargement in panel (a) is covering the letter “i” in the word “ring”. Also, use “MS & C rings” in the plural form as these are two different rings

Re: Corrected.

16- There are multiple typos in Figure S2: “polar coordinates” not “polar coordinate”, “After” not “Afer”, “LP ring &” not “LP ring&”, “Ignore” not “Ignor”. Please check this figure carefully.

Re: Corrected.

17- Figure S7 panel (b): in this table “RMSD” is misspelled.

Re: Corrected.

18- Figure S9 panel (b): add (N.F.) to the column of V? I presume you did not see filaments here as FliC and FlgE were not secreted to the culture?

Re: We added N.F. to V. Yes, we did not see the filaments because the P ring formation was disturbed and FliC and FlgE could not pass the outer membrane.

19- Figure S10 d: “predicted Δ 54-57” not “predected Δ 54-57”.

Re: Corrected.

**Response to the reviewers' comments:
To Reviewer #3:**

The authors have done a very good job of revising the manuscript. All of my comments have been addressed comprehensively.

Thank you very much. We really appreciate it.